# Tissue acidosis induces neuronal necroptosis via ASIC1a channel independent of its ionic conduction

**Yi-Zhi Wang[1†], Jing-Jing Wang[1†], Yu Huang[1], Fan Liu[1], Wei-Zheng Zeng[1], Ying Li[1], Zhi-Gang Xiong[2], Michael X Zhu[3], Tian-Le Xu[1]\***

[1]Discipline of Neuroscience, Department of Anatomy, Histology and Embryology, Collaborative Innovation Center for Brain Science, Shanghai Key Laboratory for Tumor Microenvironment and Inflammation, Shanghai Jiao Tong University School of Medicine, Shanghai, China; [2]Neuroscience Institute, Morehouse School of Medicine, Atlanta, United States; [3]Department of Integrative Biology and Pharmacology, The University of Texas Health Science Center at Houston, Houston, United States

**Abstract** Acidotoxicity is common among neurological disorders, such as ischemic stroke. Traditionally, $Ca^{2+}$ influx via homomeric acid-sensing ion channel 1a (ASIC1a) was considered to be the leading cause of ischemic acidotoxicity. Here we show that extracellular protons trigger a novel form of neuronal necroptosis via ASIC1a, but independent of its ion-conducting function. We identified serine/threonine kinase receptor interaction protein 1 (RIP1) as a critical component of this form of neuronal necroptosis. Acid stimulation recruits RIP1 to the ASIC1a C-terminus, causing RIP1 phosphorylation and subsequent neuronal death. In a mouse model of focal ischemia, middle cerebral artery occlusion causes ASIC1a-RIP1 association and RIP1 phosphorylation in affected brain areas. Deletion of the *Asic1a* gene significantly prevents RIP1 phosphorylation and brain damage, suggesting ASIC1a-mediated RIP1 activation has an important role in ischemic neuronal injury. Our findings indicate that extracellular protons function as a novel endogenous ligand that triggers neuronal necroptosis during ischemia via ASIC1a independent of its channel function.

**\*For correspondence:**
xu-happiness@shsmu.edu.cn

[†]These authors contributed equally to this work

**Reviewing editor**: Indira M Raman, Northwestern University, United States

## Introduction

Acidosis occurs commonly in a variety of neurological disorders and is a main contributing factor to neural injury (*Xiong et al., 2004*; *Wemmie et al., 2013*; *Friese et al., 2007*; *Vergo et al., 2011*). For example, ischemic stroke causes pronounced brain acidosis (~pH 6.0), the treatment of which with $NaHCO_3$ resulted in a significant reduction in the infract volume (*Pignataro et al., 2007*). Interestingly, a similar neuroprotective effect was achieved via either pharmacological intervention or genetic deletion of acid-sensing ion channel 1a (ASIC1a), suggesting a critical role for ASIC1a in mediating ischemic acidotoxicity (*Xiong et al., 2004*; *Gao et al., 2005*; *Pignataro et al., 2007*). ASIC1a belongs to the $H^+$-gated subgroup of the degenerin/epithelial $Na^+$ channel (DEG/ENaC) family of non-selective cation channels, widely expressed in central (CNS) and peripheral (PNS) nervous systems (*Wemmie et al., 2006*, *2013*). To date, besides ischemic stroke, accumulating evidence from cell/animal models shows that ASIC1a is an effective molecular target for mitigating acid-induced neural damage in many other diseases including multiple sclerosis, Huntington's disease, and Parkinson's disease (*Wemmie et al., 2013*). Thus, these previous findings strongly suggest that ASIC1a is the key extracellular proton receptor in neurons and the main mediator of acid-induced neuronal death. As such, ASIC1a may be a potential broad-spectrum therapeutic target in many neurological disorders (*Xiong et al., 2004*; *Wemmie et al., 2006*, *2013*).

**eLife digest** What happens in the minutes and hours after a stroke can determine how much brain damage occurs. In some types of stroke, a blood clot cuts off the blood supply to part of the brain, depriving the brain cells of oxygen and other nutrients, including glucose. One of the consequences is that the blood-starved brain becomes more acidic, which triggers cell death. Protecting brain cells from acidity-induced death could therefore reduce the damage caused by a stroke, and may also be an effective treatment for other brain disorders that involve increased brain acidity, like multiple sclerosis and Huntington's disease.

To create such treatments, researchers must first understand how increased acidity in the brain triggers cell death. A protein called the acid-sensing ion channel 1a (ASIC1a) is thought to contribute to acid-induced cell death by allowing calcium to flow into cells. However, this increased flow of calcium occurs only briefly (for seconds) in response to increased acidity, which cannot explain why the severity of cell death strongly depends on the length of increased brain acidity that lasts for hours during stroke.

Wang, Wang et al. now show that while ASIC1a is essential for acid-induced brain cell death, this is not because it allows calcium to enter cells. Instead, when acid levels increase, a protein called RIP1 comes to bind to one end of the ASIC1a protein. This causes the addition of a phosphate tag to RIP1, an important cellular process well known to cause the cell to die.

Wang, Wang et al. found that in mice genetically engineered to lack ASIC1a, the phosphate tag is not added to RIP1, and the brain cells survive the increased acidity caused by stroke. This suggests that preventing ASIC1a and RIP1 from interacting could be a new way to protect brain cells from the increased acidity caused by brain diseases.

However, despite the strong appreciation of its importance, the mechanism underlying ASIC1a-mediated acidic neuronal death remains poorly understood. Traditionally, $Ca^{2+}$ influx through homomeric ASIC1a channels has been considered to be the main cause of acidotoxicity (*Xiong et al., 2004*; *Yermolaieva et al., 2004*). Although intracellular $Ca^{2+}$ elevation is generally responsible for cell death mediated by $Ca^{2+}$-permeable channels, for example, NMDA receptors (*Dong et al., 2006*; *Lai et al., 2014*), this hypothesis is incompatible with some intrinsic properties of ASIC1a channels. First, homomeric ASIC1a channels are completely desensitized after just a few seconds of continued exposure to extracellular acid, a phenomenon termed steady-state desensitization (*Krishtal, 2003*; *Duan et al., 2011*). Second, compared to many other $Ca^{2+}$-permeable channels, such as NMDA receptors, the $Ca^{2+}$ permeability of ASIC1a channels is relatively small (*Samways et al., 2009*; *Wang and Xu, 2011*). Thus, under pathological conditions with persistent acidosis (e.g., for hours), $Ca^{2+}$ influx through ASIC1a channels should only occur in the first few seconds at the onset of acidosis, which would generate a negligible increase in intracellular $Ca^{2+}$ (*Samways et al., 2009*). It is unlikely that such a small change in the intracellular $Ca^{2+}$ level can fully account for the dramatic neuronal damage mediated by ASIC1a under pathological conditions. Thus, additional mechanism(s) must be at work for ASIC1a-mediated neuronal death.

Necrosis is an important form of cell death in development and diseases (*Syntichaki and Tavernarakis, 2003*; *Vandenabeele et al., 2010*). Traditionally, necrosis was considered to be an accidental, uncontrolled form of cell death (*Syntichaki and Tavernarakis, 2003*). However, accumulating evidence now suggests that necrotic cell death may be accomplished by a set of signal transduction pathways and execution mechanisms (thus termed necroptosis) (*Vandenabeele et al., 2010*; *Linkermann and Green, 2014*). Recently, the serine/threonine kinase receptor interaction protein 1 (RIP1) was identified as the crucial mediator of this process, with RIP1 phosphorylation being the key step in necroptosis (*Degterev et al., 2005*; *Christofferson et al., 2014*). Blockade of RIP1 phosphorylation by the specific inhibitor, necrostatin-1 (Nec-1), inhibited necroptosis (*Degterev et al., 2005*; *Christofferson et al., 2014*).

Necroptosis plays a critical role in many pathophysiological processes, such as ischemic injury and viral infection (*Vandenabeele et al., 2010*; *Christofferson et al., 2014*; *Linkermann and Green, 2014*). In most studies, necroptosis was mainly induced by the activation of death receptors (DRs), for example, tumor necrosis factor-α (TNF-α) receptor, in the absence of caspase activation (*He et al., 2009*;

*Vandenabeele et al., 2010*). Other endogenous initiators of necroptosis remain largely unknown. Here we report that extracellular protons trigger a novel form of necroptosis in neurons via ASIC1a, but independent of its ion-conducting function. Using RNA interference and pharmacological blockade, we identified RIP1 as a critical component in acid-induced neuronal death. RIP1 is recruited to ASIC1a within 30 min of acid stimulation, which causes RIP1 phosphorylation and triggers the downstream death events. Similarly, RIP1 became physically associated with ASIC1a and hyperphosphorylated in response to middle cerebral artery occlusion (MCAO) in mice. The enhanced RIP1 phosphorylation in response to either ischemia or acidosis was undetected in neurons from *Asic1a$^{-/-}$* mice, demonstrating the involvement of ASIC1a-mediated RIP1 activation in ischemic brain injury.

## Results

### Neurons undergo necrotic death in response to acidosis

According to morphological appearance, cell death can be divided into apoptotic and necrotic death (*Kroemer et al., 2009*). Either death form represents a specific set of signaling pathways and biochemical/cellular processes (*Kroemer et al., 2009*). In order to classify acid-induced neuronal death, we first examined the morphological changes of cultured mouse cortical neurons exposed to acidosis using electron microscopy (EM). While most neurons treated with a pH 7.4 solution (*Figure 1—figure supplement 1A*, upper panel) showed normal cellular morphology (*Figure 1A1*, left panel; *Figure 1A2*, upper panel), those treated with a pH 6.0 solution (1 hr treatment and 24 hr recovery in normal culture medium, *Figure 1—figure supplement 1A*, middle panel) displayed a typical necrotic phenotype (*Kroemer et al., 2009*), including plasma membrane rupture, organelle swelling, and cell lysis (*Figure 1A1*, middle and right panels; *Figure 1A2*, lower panel). No obvious apoptotic morphological change was observed, based on comparison with staurosporine-treated neurons (data not shown).

Because our culture system lacks phagocytes, we measured caspase activities to evaluate the possible involvement of secondary necrosis following apoptosis (*Berghe et al., 2010*). Although the activity of caspase 8 was moderately increased 4 and 8 hr after the pH 6.0 treatment (*Figure 1—figure supplement 1B*), that of executive apoptosis effectors, caspase 3/7, did not increase (*Figure 1—figure supplement 1C-E*). Rather, caspase 3/7 activities decreased slightly at 4 hr (*Figure 1—figure supplement 1C*) and the cleavage of caspase 3 (activated caspase 3) (*Figure 1—figure supplement 1E*) was undetectable by Western blotting. These data, together with the result that pan caspase inhibitor z-VAD-fmk failed to block pH 6.0-induced neuronal death (*Figure 1B*, *Figure 1—figure supplement 1F*), suggest that acid triggers necrotic neuronal death but not apoptosis under the current experimental conditions.

### Acid-induced neuronal death is both ASIC1a- and RIP1-dependent

We then adopted a modulatory profiling strategy (*Wolpaw et al., 2011*) to explore the molecular mechanism(s) underlying acid-induced neuronal death. Based on the literature (*Berghe et al., 2010*; *Wolpaw et al., 2011*), we tested the effect of eight death modulators, including scavengers of reactive oxygen species (ROS) and inhibitors of Ca$^{2+}$ signaling, protein synthesis, proteases, ASIC1a, and the main necroptosis mediator, RIP1, all with 30 min pretreatment and then co-incubation with the pH 6.0 solution (*Figure 1—figure supplement 1A*, lower panel). Cell viability was assessed by the Cell Titer Blue (CTB) assay. As reported previously (*Xiong et al., 2004*), the ASIC1a specific inhibitor, psalmotoxin (PcTX1), attenuated acid-induced neuronal death. To our surprise, however, inhibiting RIP1 phosphorylation (*Figure 2A,B*) by Nec-1 (*Degterev et al., 2005*) also resulted in a similar protective effect (IC$_{50}$, 12.6 μM, *Figure 1B*; CTB assay). Unlike PcTX1, up to 30 min pretreatment with Nec-1 did not affect the acid (pH 6.0)-evoked currents ($I_{6.0}$) in the neurons (*Figure 1—figure supplement 2A*).

As an alternative method for examining necrotic neuronal death, neurons were stained with propidium iodide (PI); the pH 6.0 treatment dramatically increased the number of PI-positive neurons. This effect was also suppressed by Nec-1 (*Figure 1C1,C2*); acid-treated neurons retained their healthy appearance of cell bodies and neurites in the presence of Nec-1 (*Figure 1C1*, bottom panel, DIC images), suggesting that neuronal functions may be preserved. Additionally, acid (pH 6.0)-induced neuronal damage was detected using the lactate dehydrogenase (LDH) assay, which, unlike the CTB assay, is independent of mitochondrial metabolism, and the effect was inhibited by PcTX1, Nec-1, and 7-Cl-O-Nec-1 (Nec-1s), a derivative with improved specificity for RIP1 (*Takahashi et al., 2012*)

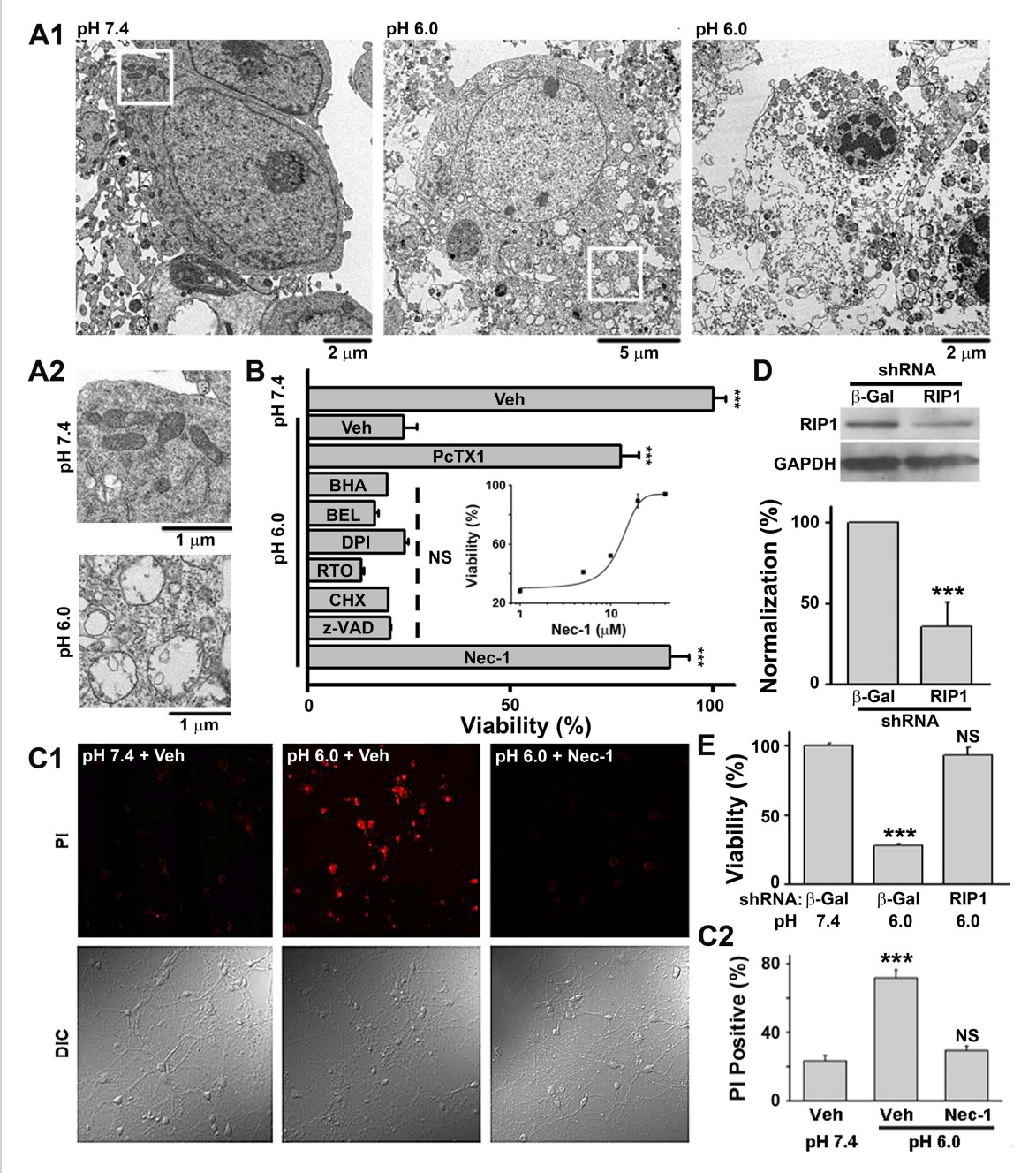

**Figure 1**. Acid (pH 6.0) induces RIP1-dependent necrotic cell death in cultured mouse cortical neurons. (**A1**) Electron microscopy images of neurons treated with pH 7.4 (left) or pH 6.0 solution (middle and right). Of 57 cells counted in the pH 6.0-treated samples, 47 showed morphology similar to that shown in the middle and right panels. For pH 7.4-treated samples, the majority of cells had a similar morphology to that shown in the left panel; only 3 out of the 41 cells examined showed morphology that resembled that in the middle panel. (**A2**) Enlarged images from the white boxes in **A1** showing swelling of organelles in pH 6.0- but not pH 7.4-treated neurons. (**B**) PcTX1 (10 nM) and Nec-1 (20 µM), but not BHA (100 µM), BEL (30 µM), DPI (15 µM), RTO (25 µM), CHX (100 µM), or z-VAD-fmk (10 µM), rescued cells from acid-induced neuronal death (indicated by the dashed line) (n=4–12, ***p<0.001; NS, no statistical significance, vs vehicle (Veh) at pH 6.0). Inset: dose-dependence of the rescue by Nec-1 (CTB assay, n=3–4). (**C1**) Rescue from acid-induced neuronal death by 20 µM Nec-1 (propidium iodide [PI] staining assay). (**C2**) Summary data for **C1**. At least 200 neurons were counted for each condition (***p<0.001; NS, no statistical significance, vs Veh at pH 7.4). (**D**) Knockdown efficiency of RIP1 shRNA as determined by Western blotting (***p<0.001, vs β-Gal). (**E**) Rescue of acid-induced neuronal death by RIP1 shRNA (CTB assay, n=3, ***p<0.001; NS, no statistical significance, vs β-Gal at pH 7.4).

*Figure 1. continued on next page*

*Figure 1. Continued*

The following figure supplements are available for figure 1:

**Figure supplement 1**. Acid (pH 6.0) treatment does not induce caspase 3/7 activation in cultured mouse cortical neurons.
**Figure supplement 2**. RIP1 mediates ASIC1a-dependent acid-induced neuronal death in cultured mouse cortical neurons.
**Figure supplement 3**. Acid (pH 6.0) treatment does not induce reactive oxygen species (ROS) production in cultured mouse cortical neurons.

(*Figure 1—figure supplement 2B*). Importantly, drug pretreatment was necessary for neuronal protection, as co-applied Nec-1 or Nec-1s failed to inhibit acidic neuronal death (*Figure 1—figure supplement 2C*), suggesting a key role for RIP1 at the onset of acidotoxicity. Consistent with the pharmacological intervention, RNA interference of endogenous RIP1 protected neurons from acid-induced death (*Figure 1D,E*). In addition, geldanamycin (GA) treatment, which greatly reduced RIP1 expression level (*Vanden Berghe et al., 2003*), also significantly suppressed acid-induced cell death (*Figure 1—figure supplement 2D*).

A previous report showed that inhibitors of acid-induced neuronal death also suppressed ROS generation (*Liu et al., 2009*), implying a potential pro-necrotic effect of ROS (*Vandenabeele et al., 2010*) on acid-induced necrosis. However, the pH 6.0 treatment resulted in no obvious change in ROS levels in the neurons (*Figure 1—figure supplement 3*) and neither scavenging ROS by anti-oxidant butylated hydroxyanisole (BHA) nor blockade of ROS production by rotenone (RTO) or diphenylene iodonium (DPI) rescued neurons from acid-induced death (*Figure 1B*). These results indicate that ROS production is not necessary for acid-induced cell death, consistent with the previous finding (*Xiong et al., 2004*). Also, earlier reports showed that extracellular protons led to intracellular $Ca^{2+}$ elevation (*Xiong et al., 2004*; *Yermolaieva et al., 2004*), which is one of the major causes of conventional necrotic cell death. Besides enhancing mitochondrial ROS production through activation of key enzymes of the Krebs cycle, $Ca^{2+}$ also triggers cytosolic phospholipase $A_2$ ($cPLA_2$)-mediated necrosis (*Vandenabeele et al., 2010*). However, a $cPLA_2$ inhibitor, bromoenol lactone (BEL), failed to protect neurons from acid-induced death (*Figure 1B*). Removing $Ca^{2+}$ from the treatment solution also had no clear protective effect in the present study (*Figure 3D*). The above results, thus, suggest that acid-induced neuronal death involves a form of RIP1-mediated necrosis which is independent of ROS and $Ca^{2+}$. Consistent with this idea, recent evidence showed that RIP1 was not involved in certain forms of intrinsic necrosis induced by ROS generators or $Ca^{2+}$ ionophores (*Sun et al., 2012*; *Wang et al., 2012*).

## Activation of ASIC1a channels leads to RIP1 phosphorylation

RIP1 is a crucial mediator in many forms of necrotic cell death and its inhibition by Nec-1 (*Degterev et al., 2005*, *2008*) is neuroprotective in rodent disease models including ischemic stroke (*Cho et al., 2009*; *He et al., 2009*; *Zhang et al., 2009*). To test whether acid causes RIP1 activation in neurons, we measured RIP1 phosphorylation, a common signature for RIP1-mediated necrotic death, using two methods: $^{32}P$ incorporation and direct detection of phosphorylated RIP1 by an anti-phospho Ser/Thr antibody from anti-RIP1 immunoprecipitated samples. Both methods showed significant increases in RIP1 phosphorylation upon 30 min treatment with the pH 6.0 solution, which were suppressed by Nec-1 (*Figure 2A,B*). Consistent with ASIC1a being critical for necrotic cell death and upstream of RIP1 activation, acid-induced RIP1 phosphorylation was inhibited by PcTX1 (*Figure 2B*) and was undetectable in neurons from $Asic1a^{-/-}$ mice (*Figure 2C*).

Intracellular acidification, which can be induced by extracellular acidosis (*Figure 2D1,D2*), modulates many biochemical functions including kinase activities (*Kraut and Madias, 2010*). To rule out the possibility that acid-induced RIP1 phosphorylation resulted from intracellular acidification, we performed an in vitro cell-free phosphorylation assay and found that reducing the pH of the reaction solution to 6.0 did not alter RIP1 phosphorylation (*Figure 2E*). Moreover, despite similar extracellular acid-induced intracellular acidification between neurons from $Asic1a^{+/+}$ (wild type, WT) and $Asic1a^{-/-}$ mice (*Figure 2D1,D2*), only WT, but not $Asic1a^{-/-}$, neurons showed enhanced RIP1 phosphorylation in response to the extracellular pH reduction (*Figure 2C*), suggesting that the presence of ASIC1a proteins rather than intracellular acidification was necessary for such a response. Interestingly, the

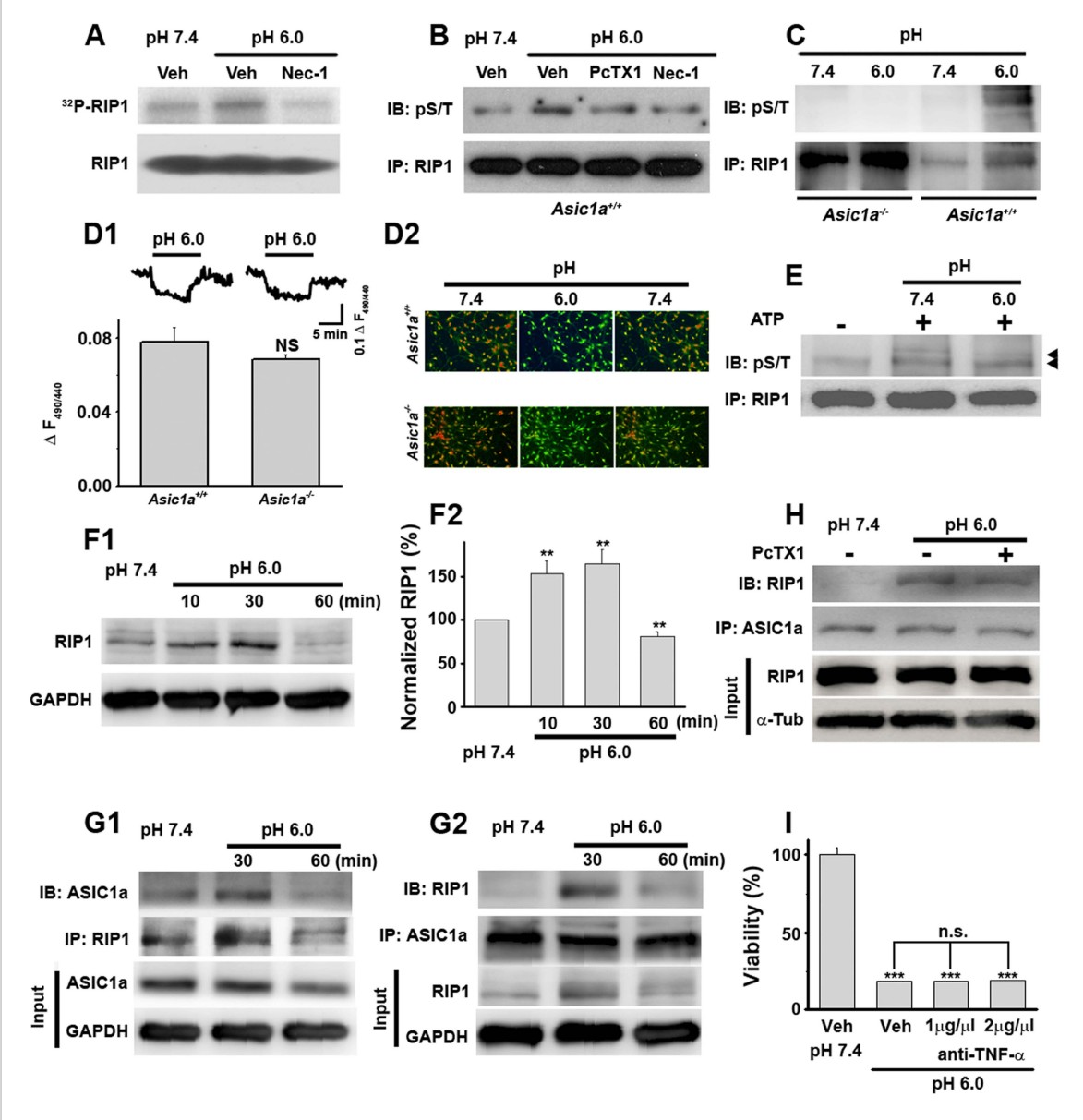

**Figure 2**. Acid (pH 6.0) induces RIP1 phosphorylation and physical association between RIP1 and ASIC1a. (**A**) Acid-induced phosphorylation of RIP1 and its inhibition by 40 μM Nec-1, as measured by $^{32}$P incorporation. (**B**) Acid-induced phosphorylation of RIP1 and its inhibition by Nec-1 (40 μM) and PcTX1 (50 nM), detected using the anti-phospho S/T antibody. (**C**) Acid failed to induce RIP1 phosphorylation in neurons from *Asic1a*$^{-/-}$ mice. Five-fold more proteins were loaded for *Asic1a*$^{-/-}$ samples than for *Asic1a*$^{+/+}$ samples. (**D1, D2**) Time courses (**D1**) and representative ratio images (**D2**) of intracellular acidification of cultured mouse cortical neurons from *Asic1a*$^{+/+}$ and *Asic1a*$^{-/-}$ mice in response to extracellular pH decrease from 7.4 to 6.0, monitored using BCECF (n=30 for each genotype, peak changes summarized in **D1**). (**E**) In vitro RIP1 phosphorylation assay in pH 7.4 and pH 6.0 reaction solutions. Bands for phosphorylated RIP1 are indicated by the arrowheads. (**F1, F2**) Acid altered RIP1 expression levels with time. Shown are representative blots (**F1**) and summary data (**F2**) (n=5, \*\*p<0.01 vs pH 7.4, by paired *t* test). (**G1, G2**) pH 6.0 treatment caused association of RIP1 with ASIC1a (**G1**, IP (immunoprecipitation) RIP1, IB (immunoblotting) ASIC1a; **G2**, IP ASIC1a, IB RIP1). (**H**) PcTX1 disrupted acid-induced ASIC1a–RIP1 association. (**I**) TNF-α neutralizing antibody (1 μg/μl and 2 μg/μl) failed to rescue pH 6.0 solution-induced neuronal death (CTB assay, n=3, \*\*\*p<0.001 vs vehicle (Veh) at pH 7.4; n.s., no statistical significance, vs Veh at pH 6.0).

The following figure supplement is available for figure 2:

**Figure supplement 1**. Controls for *Figure 2G,H,I*.

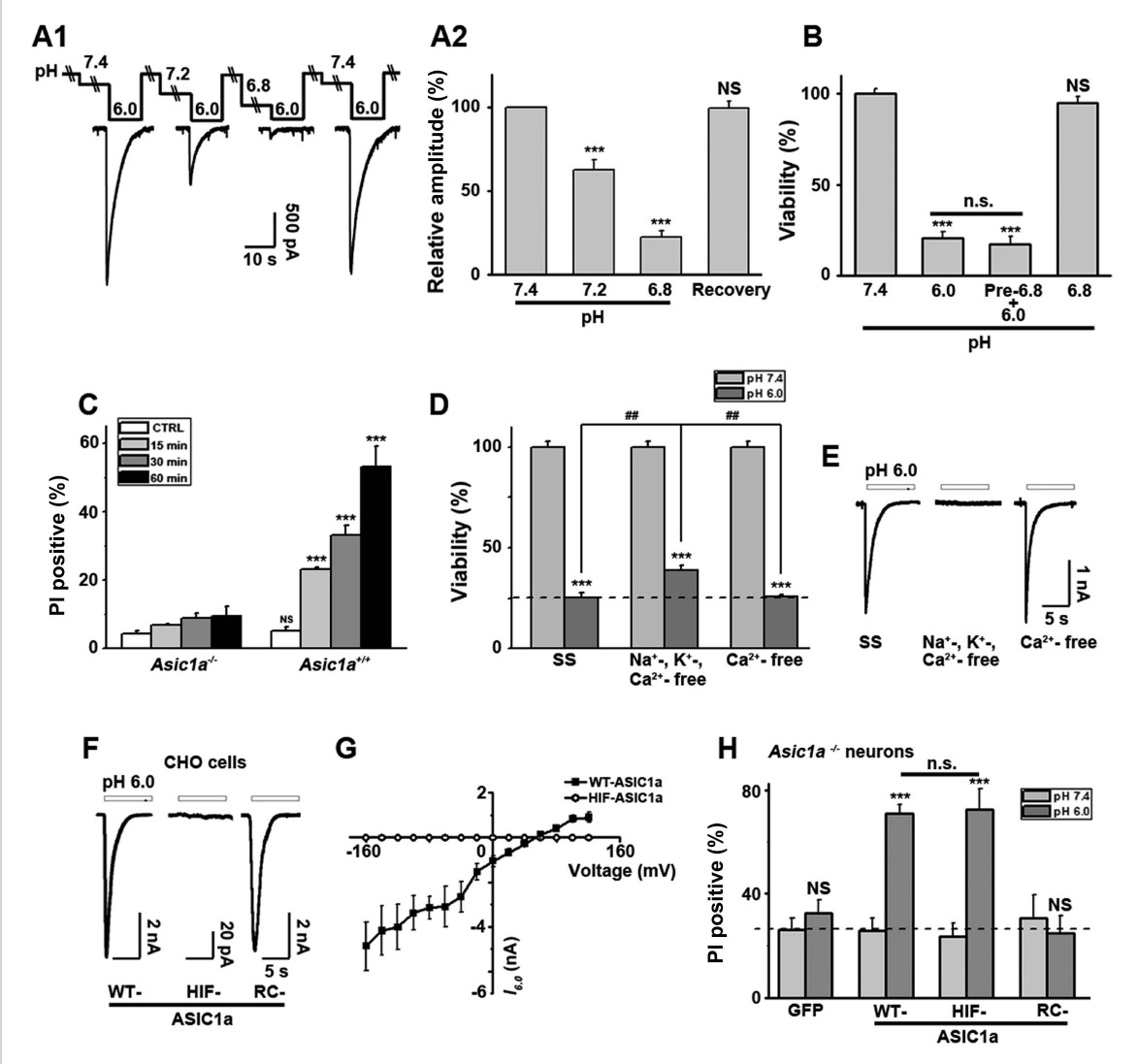

**Figure 3**. Ion-conducting function is not necessary for ASIC1a-mediated neuronal death. (**A1**, **A2**) Two-minute pretreatment with mild acidification greatly suppressed $I_{6.0}$ in cultured mouse cortical neurons. Shown are representative current traces at −60 mV (**A1**) and summary data for peak currents (**A2**) (n=4–6, ***p<0.001; NS, no statistical significance, vs pH 7.4). (**B**) Pretreatment with pH 6.8 (Pre-6.8) for 10 min failed to prevent acid-induced neuronal death (CTB assay, n=4–6, ***p<0.001, NS, no statistical significance, vs pH 7.4; n.s., no statistical significance, vs pH 6.0). Note: treatment with the pH 6.8 solution for 1 hr only did not alter neuronal viability. (**C**) Summary of propidium iodide (PI)-positive neurons for $Asic1a^{+/+}$ and $Asic1a^{-/-}$ cultures treated with pH 7.4 (CTRL) or pH 6.0 solutions for 15, 30, and 60 min. Representative images are shown in ***Figure 3—figure supplement 1B***. At least 400 neurons were counted for each condition (***p<0.001; NS, no statistical significance, vs corresponding $Asic1a^{-/-}$ cultures). (**D**) Neuronal death induced by 1 hr treatment with the pH 6.0 solution in normal (SS, standard external solution), $Na^+$-, $K^+$-, $Ca^{2+}$-free (NMDG replacement), and $Ca^{2+}$-free conditions (CTB assay, n=3, ##p<0.01 vs $Na^+$-, $K^+$-, $Ca^{2+}$-free in pH 6.0; ***p<0.001 vs pH 7.4 under the same cation conditions). Dashed line, pH 6.0 solution-induced neuronal death under normal SS condition. (**E**) Representative traces of $I_{6.0}$ for cortical neurons in normal (SS), $Na^+$-, $K^+$-, $Ca^{2+}$-free and $Ca^{2+}$-free conditions. (**F**) Representative traces of $I_{6.0}$ for wild type (WT) ASIC1a and its HIF and RC mutants expressed in CHO cells. (**G**) Current–voltage relationship of WT-ASIC1a (filled squares) and HIF-ASIC1a (open circles) in response to the pH 6.0 solution. Note: no current was induced by pH 6.0 in CHO cells that expressed HIF-ASIC1a at a broad range of holding voltages. (**H**) Summary data of cell death induced by 1 hr pH 6.0 solution treatment in $Asic1a^{-/-}$ neurons expressing GFP vector, WT-ASIC1a, and its HIF and RC mutants, based on PI staining of GFP-labeled neurons (see ***Figure 3—figure supplement 6*** for representative images, n=100 for each condition, ***p<0.001; NS, no statistical significance, vs pH 7.4 of the corresponding transfection; n.s., no statistical significance, vs WT-ASIC1a).

The following figure supplements are available for figure 3:

**Figure supplement 1**. Time-dependence of acid-induced neuronal death.

*Figure 3. continued on next page*

*Figure 3. Continued*

**Figure supplement 2**. Fast perfusion is necessary for acid to induce $[Ca^{2+}]_i$ elevation in ASIC1a-expressing neurons.

**Figure supplement 3**. Validation of the Y-tube apparatus at fast and slow perfusion rates.

**Figure supplement 4**. Acid-induced neuronal death is independent of $Ca^{2+}$-flux and ionic currents via ASIC1a.

**Figure supplement 5**. HIF and RC mutants of ASIC1a are normally expressed on the plasma membrane.

**Figure supplement 6**. Expression of WT-ASIC1a and HIF-ASIC1a, but not RC-ASIC1a, in *Asic1a⁻/⁻* neurons resulted in acid-induced death.

**Figure supplement 7**. CHO cell death induced by acid (pH 6.0) is dependent on ASIC1 and involves recruitment of RIP1.

total protein level of RIP1 was also increased in the first 30 min of acidosis treatment and then declined to ~80% of the basal level at 60 min (*Figure 2F1,F2*). Although a decrease in RIP1 level was previously observed in other forms of cell death (*Lin et al., 1999*; *Van de Walle et al., 2010*), its role in acid-induced neuronal death requires further exploration. Furthermore and possibly related to the acid-induced increase in RIP1 phosphorylation, we found that RIP1 was recruited to ASIC1a in mouse cortical neurons after 30 min treatment with the pH 6.0 solution (*Figure 2G1,G2*, *Figure 2—figure supplement 1A1,A2*) and that this effect was suppressed by PcTX1 (*Figure 2H*). Taken together, these results suggest that acid-induced neuronal death is a programmed form of RIP1-dependent necrosis, or necroptosis (*Degterev et al., 2005*; *Linkermann and Green, 2014*), which is dependent on the physical association of RIP1 with ASIC1a.

## ASIC1a-mediated neuronal necroptosis is independent of other death receptors

Nearly all forms of previously reported necroptosis are induced by death receptor (DR) ligands, such as TNF-α, particularly in the presence of caspase inhibitors (*Vandenabeele et al., 2010*). Recent studies showed that, under certain conditions, autocrine production of TNF-α could lead to RIP1-mediated necroptosis (*Biton and Ashkenazi, 2011*; *Wu et al., 2011*). To evaluate the possibility that acid-induced necroptosis arose from autocrine production of cytokines, we first tested whether de novo protein synthesis was required for cell death and found that the protein synthesis inhibitor cycloheximide (CHX) failed to protect neurons from acid-induced death (*Figure 1B*). Second, we used the TNF-α neutralizing antibody but found that it failed to rescue neurons from acid-induced death, while it was effective in protecting neurons from TNF-α-induced death (*Figure 2I*, *Figure 2—figure supplement 1B*). Therefore, it is unlikely that any pro-necrotic protein production is involved in acid-induced necrotic cell death.

## Steady-state desensitized ASIC1a channels are able to mediate acidotoxicity

Interestingly, Nec-1 significantly reduced acid-induced neuronal death without affecting $I_{6.0}$ (*Figure 1—figure supplement 2A*), suggesting that the ion-conducting function of ASIC1a was perhaps unrelated to RIP1-dependent acidotoxicity. To test this hypothesis, we examined the steady-state desensitization of proton-evoked currents in cortical neurons by moderate pH decreases, that is, to pH 7.2 or 6.8. Just 2 min superfusion with the pH 7.2 or pH 6.8 solution drastically reduced $I_{6.0}$ by approximately 60% or 90%, respectively (*Figure 3A1,A2*). Therefore, ASIC1a channels in these neurons desensitized, leaving nearly no ionic flux during continued exposure to acid (>2 min). In contrast, pretreatment of neurons with pH 6.8 solution failed to inhibit acid-induced neuronal death (*Figure 3B*, *Figure 3—figure supplement 1A*), indicating that ASIC1a channels were able to mediate acidotoxicity even under steady-state desensitized conditions.

As it is an ion channel, the ion-conducting function is generally considered a main consequence of ASIC1a activation. However, it is rather paradoxical that whereas the severity of acid-induced

neuronal death correlates positively with the duration of acid treatment (*Xiong et al., 2004*; *Duan et al., 2011*), the ionic currents mediated by homomeric ASIC1a channels only last for seconds due to the complete steady-state channel desensitization at acidic pH, at least in in vitro conditions (*Krishtal, 2003*; *Duan et al., 2011*). To examine the time dependence on acidosis of neuronal necrosis, we treated the cortical neurons with the pH 6.0 solution for different durations (0, 15, 30, and 60 min) and then returned them to the normal culture medium for 24 hr before staining with PI. As shown in *Figure 3C* and *Figure 3—figure supplement 1B*, the number of PI-positive neurons continued to increase with increasing durations of the pH 6.0 treatment in cultures prepared from $Asic1a^{+/+}$ mice, indicating that despite the nearly complete channel desensitization throughout the major part (except for the first few seconds) of the 1 hr treatment, acid continued to exert an effect on neuronal death. On the other hand, consistent with previous findings (*Xiong et al., 2004*; *Duan et al., 2011*), acid treatment failed to induce significant neuronal death in cultures prepared from $Asic1a^{-/-}$ mice for all treatment durations (*Figure 3C*, *Figure 3—figure supplement 1B*), indicating that despite the lack of proton-induced current, the neuronal death induced by persistent acidosis was still dependent on the presence of ASIC1a. Therefore, although ASIC1a expression was necessary, continued ASIC1a currents appeared not critical for acid-induced cell death.

## ASIC1a-mediated neuronal death is not dependent on its ionic conduction

$Ca^{2+}$ influx via homomeric ASIC1a channels has been considered to play a key role in acidic neuronal death (*Xiong et al., 2004*; *Yermolaieva et al., 2004*; *Wang and Xu, 2011*). Consistent with previous studies, application of the pH 6.0 solution to the cultured mouse cortical neurons elicited a rise in intracellular $Ca^{2+}$ concentration ($[Ca^{2+}]_i$), which was significantly reduced following inhibition of ionotropic glutamate receptors and voltage-gated $Na^+$ and $Ca^{2+}$ channels, while the remaining response was largely blocked by PcTX1 (*Figure 3—figure supplement 2A1,A2*). The $Ca^{2+}$ response was not detected in neurons from $Asic1a^{-/-}$ mice (*Figure 3—figure supplement 2B1*), but restored with the transient expression of ASIC1a cDNA in these neurons (*Figure 3—figure supplement 2B2,B4*). However, the $Ca^{2+}$ response was only seen with fast focal application of the pH 6.0 solution (~50 µl/min). When the perfusion rate was decreased by approximately threefold (~15 µl/min), the $[Ca^{2+}]_i$ rise became very shallow or nearly undetectable (*Figure 3—figure supplement 2B5,B6*). This sensitivity to the rate of acidification is likely related to the fast desensitization of acid-induced ASIC1a current, because the acid-evoked activation of TRPV1 (either pH 6.0 or 5.0), which did not desensitize, was unaffected by the perfusion rate (*Figure 3—figure supplement 3A,B,D*). For fast desensitizing channels, it requires simultaneous activation of the majority of channels in the entire cell in order to give rise to robust whole-cell currents (*Figure 3—figure supplement 3C*) and $[Ca^{2+}]_i$ elevation (*Figure 3—figure supplement 2*). Given that tissue acidosis occurs slowly under most pathological conditions, we reasoned that ASIC1a-mediated $[Ca^{2+}]_i$ elevation during this process must be quite small and unlikely a major cause of the acid-induced neuronal death (*Wang and Xu, 2011*). Indeed, a $Ca^{2+}$-free pH 6.0 solution (with no $Ca^{2+}$ added and the addition of 10 mM EGTA), which would not support $Ca^{2+}$ influx, induced neuronal death to a similar level as the normal $Ca^{2+}$-containing pH 6.0 solution (*Figure 3D*). Importantly, acid-induced neuronal death in the $Ca^{2+}$-free condition was also inhibited by PcTX1 and Nec-1s (*Figure 3—figure supplement 4A*), indicating the involvement of the ASIC1a-RIP pathway despite the lack of $Ca^{2+}$ influx. Notably, unlike $I_{6.0}$ in ASIC1a-transfected CHO cells (*Duan et al., 2011*), depletion of extracellular $Ca^{2+}$ did not reduce $I_{6.0}$ in cultured mouse cortical neurons (*Figure 3E*), suggestive of a different modulatory mechanism exerted on native ASIC1a by extracellular $Ca^{2+}$ under the present experimental conditions.

To further examine the contribution of ion influx to acid-induced neuronal death, we replaced $Na^+$, $K^+$, and $Ca^{2+}$ in the treatment solution with an impermanent cation, *N*-methyl-D-glucamine (NMDG), keeping osmolarity unchanged. Under these conditions, no $I_{6.0}$ was detected because of the lack of permeant ions (*Figure 3E*), but acid still resulted in marked neuronal death (*Figure 3D*). We also tested the non-specific ASIC blocker, amiloride (AMI), which acts by blocking the channel pore. Although co-administration of AMI with acid significantly inhibited $I_{6.0}$ (*Figure 3—figure supplement 4B1,B2*), it did not show neuroprotection (*Figure 3—figure supplement 4C*). These results further indicated that the ion fluxes or conducting function of ASIC1a might not be required for acid-induced neuronal death. Interestingly, if neurons were pretreated with AMI for 1 hr before acid stimulation, the acidotoxicity was largely prevented (*Figure 3—figure supplement 4C*), suggesting that pretreatment with AMI may modulate the ASIC1a channel with a different mechanism beyond the channel blockade.

## The C-terminus of ASIC1a is crucial to acid-induced neuronal death

Being an extracellular proton sensor, ASIC1a channels undergo conformational changes irrespective of the ion-conducting outcome. As such, the death pathway may only require the proton sensor function rather than ionic conduction. To test this possibility, we created two ASIC1a mutants: HIF-ASIC1a and RC-ASIC1a (*Figure 3—figure supplement 5A*). While both were expressed normally on the plasma membrane (*Figure 3—figure supplement 5B*), HIF-ASIC1a ($^{32}$HIF$^{34}$ mutated to $^{32}$AAA$^{34}$) was ion non-conducting due to pore dysfunction (*Pfister et al., 2006*) (*Figure 3F,G*). RC-ASIC1a was electrophysiologically functional as the WT channel (*Figure 3F*), but its C-terminus (R$^{462}$-C$^{526}$) was replaced by a shortened scrambled amino acid sequence, KLRILQSTVPRARDDPDLDN (*Figure 3—figure supplement 5A*). By expressing WT-ASIC1a, HIF-ASIC1a, or RC-ASIC1a in cortical neurons from *Asic1a$^{-/-}$* mice, we found that both WT-ASIC1a and the ion non-conducting HIF mutant, but not the conducting RC mutant, restored the acid-induced death in *Asic1a$^{-/-}$* neurons (*Figure 3H*, *Figure 3—figure supplement 6*), although both WT- and RC-ASIC1a, but not the HIF mutant, restored the Ca$^{2+}$ response to fast focal perfusion with the pH 6.0 solution (*Figure 3—figure supplement 2B2–B4*). Furthermore, although CHO cells expressing homomeric ASIC1a failed to yield a [Ca$^{2+}$]$_i$ rise in response to acid (*Figure 3—figure supplement 7A1,A2*), the stable expression of WT-ASIC1a and HIF-ASIC1a, but not RC-ASIC1a, in CHO cells also led to significant increases in acid-induced cell death (*Figure 3—figure supplement 7B*). Supporting a similar ASIC1a/RIP1-mediated death mechanism in CHO cells as in neurons, pH 6.0 solution treatment also caused ASIC1a–RIP1 association in CHO cells that expressed ASIC1a (*Figure 3—figure supplement 7C*).

Further supporting the importance of the ASIC1a C-terminus in acid-induced necroptosis, we found that ASIC1b, a splice variant differing from ASIC1a only at the N-terminus, also mediated acid-induced death when expressed in CHO cells (*Figure 3—figure supplement 7D*). In contrast, the expression of ASIC2a and ASIC3, which have very different C-terminal sequences from ASIC1a (*Figure 4—figure supplement 1A*), did not restore the acid-induced death of CHO cells (*Figure 3—figure supplement 7D*). Because homomeric ASIC1b is Ca$^{2+}$ impermeable (*Bassler et al., 2001*), this finding also supports the argument that Ca$^{2+}$ entry through ASIC1a is not critical for acid-induced cell death under the present experimental conditions. The above data, thus, strongly support the notion that ASIC1a-mediated neuronal death does not require the ion permeation ability of the ASIC1a channel. Rather, a death signal presumably located at the C-terminus of the channel protein and activated upon stimulation by extracellular protons might be responsible for the acid-induced cell death.

## A peptide representing the proximal C-terminus of ASIC1 mimics acidic neuronal necroptosis

Because conservation at the C-termini of ASIC isoforms is very low, the sequence alignment (*Figure 4—figure supplement 1A*) was uninformative about potential key amino acids for acidic neuronal death. We reasoned that synthetic peptides representing the critical motif(s) involved in acid-induced necroptosis might be able to mimic the action of the ASIC1a C-terminus on cell death. Therefore, we synthesized four peptides based on the mouse ASIC1a C-terminus, designated as CP-1, 2, 3, and 4 (see *Figure 4A*, *Figure 4—figure supplement 1A*). CP-1 covered a relatively less conserved region among ASIC C-termini than CP-2, CP-3, and CP-4 (*Figure 4—figure supplement 1A*). All four peptides were tagged with the TAT sequence to facilitate penetration through the plasma membrane (*Figure 4A*). Interestingly, incubation of mouse cortical neurons with CP-1 (10 μM, 24 hr), but not CP-2, CP-3, or CP-4, at the physiological pH 7.4, induced cell death (*Figure 4B*) and enhanced RIP1 phosphorylation (*Figure 4C*), which were both rescued, at least partially, by Nec-1 (*Figure 4D,E*). However, none of the peptides affected acid-induced neuronal death (data not shown) and in the presence of CP-1, the pH 6.0 solution still induced further loss of neuronal viability (*Figure 4—figure supplement 1B*). Moreover, the CP-1 peptide caused the death of CHO cells (*Figure 4—figure supplement 1C*), which do not endogenously express ASIC1a, suggesting that its toxicity does not require full-length ASIC1a. These data suggest that the proximal C-terminal region of ASIC1a included in the CP-1 peptide can induce neuronal death via activation of RIP1, mimicking acid-induced necroptosis.

## RIP1 is recruited to ASIC1a and phosphorylated in ischemic brain

The above experiments firmly established the role of ASIC1a–RIP1 coupling in the acid-induced death of cultured cortical neurons. To verify the involvement of this pathway in ischemic brain injury, which is

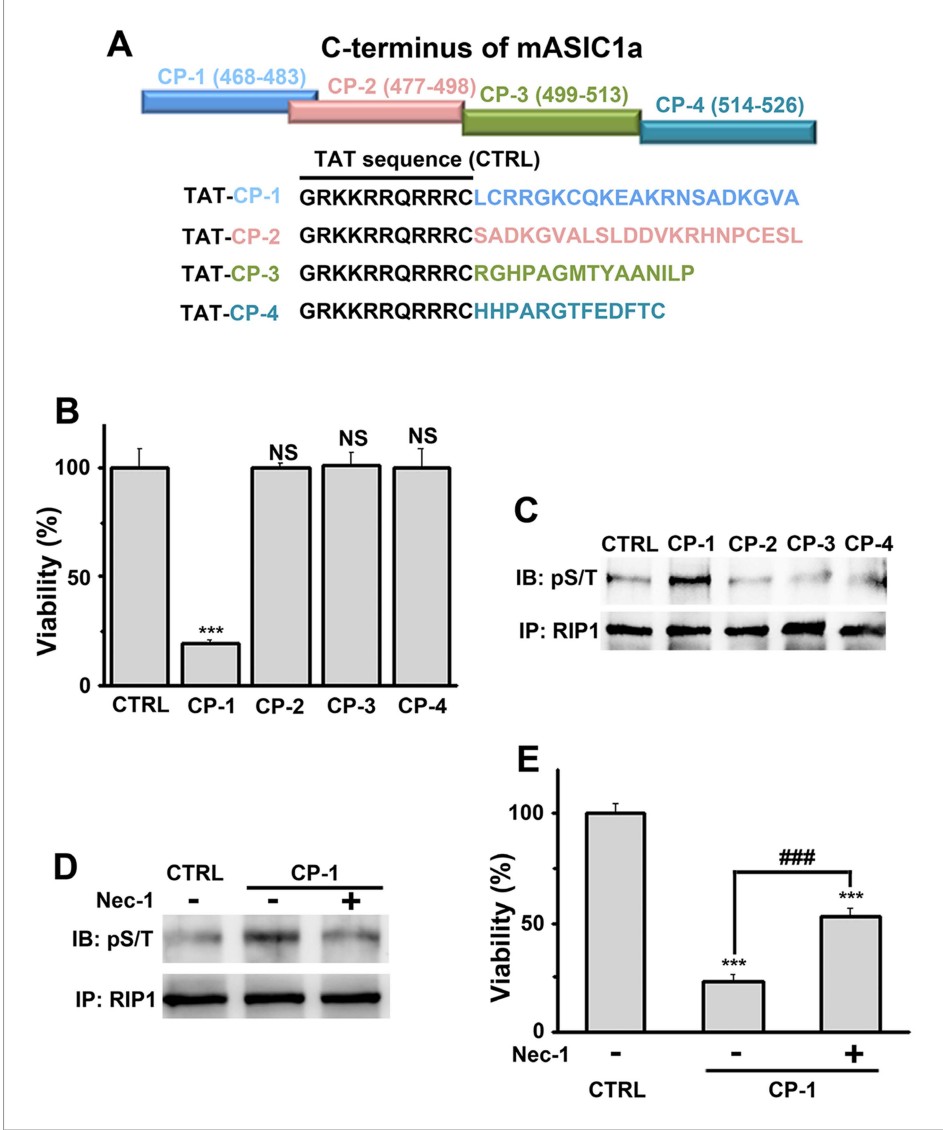

Figure 4. A peptide representing the proximal C-terminal region of ASIC1a induces RIP1 phosphorylation and neuronal death. (A) Relative positions and amino acid sequences of four peptides representing different regions of the mouse ASIC1a (mASIC1a) C-terminus. (B) TAT-tagged CP-1, but not CP-2, CP-3, or CP-4, peptide (10 μM, 24 hr) induced neuronal death at pH 7.4 (CTB assay, n=3, ***p<0.001; NS, no statistical significance, vs control [CTRL, the TAT peptide alone]). (C, D) CP-1, but not CP-2, CP-3, or CP-4, enhanced RIP1 phosphorylation (C) and the effect was blocked by Nec-1 (20 μM) (D). (E) Nec-1 (20 μM) partially rescued CP-1-induced neuronal death (CTB assay, n=3, ***p<0.001 vs CTRL; ###p<0.001 vs CP-1 alone).

The following figure supplement is available for figure 4:

Figure supplement 1. Design and characterization of ASIC1a-derived peptides.

accompanied by severe tissue acidosis, with pH values typically falling to 6.5–6.0 (*Bassler et al., 2001*; *Xiong et al., 2004*; *Linkermann and Green, 2014*), we used a transient MCAO model to mimic ischemic stroke in mice (*Figure 5A*). Supporting the role of ASIC1a in mediating RIP1 activation in ischemic brain damage, we detected a physical association between RIP1 and ASIC1a only in the ischemic hemisphere but not the control hemisphere of the same brain (*Figure 5B*). Importantly, 30 min MCAO, which was too short to cause obvious brain damage as shown by TTL staining (*Figure 5C*), was sufficient to induce the ASIC1a-RIP1 association. Such association persisted for up to

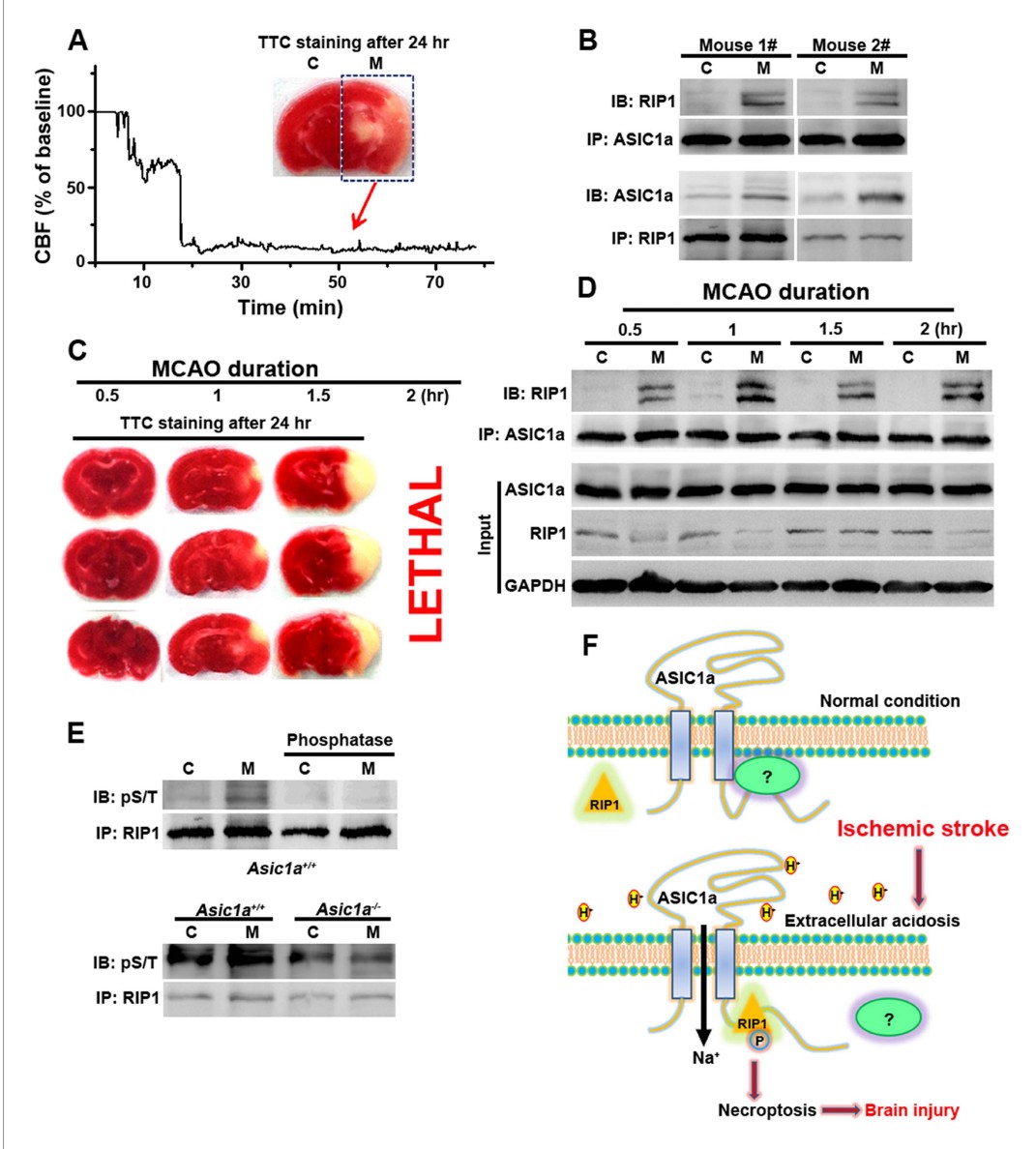

**Figure 5**. RIP1 is recruited to ASIC1a in ischemic brain. (**A**) Cerebral blood flow (CBF) of ischemic brain hemisphere before and during middle cerebral artery occlusion (MCAO, arrow) was monitored by transcranial laser Doppler. C, control hemisphere; M, MCAO hemisphere. (**B**) One-hour MCAO treatment caused association of RIP1 with ASIC1a (upper, IP: ASIC1a, IB: RIP1; lower, IP: RIP1, IB: ASIC1a). Both bands in IB: RIP1 represent RIP1; the upper band may represent phospho-RIP1 or ubiquitinated-RIP1 as shown in previous studies (*Cho et al., 2009*; *He et al., 2009*). (**C**) TTC-staining of ischemic brain slices after 0.5, 1, and 1.5 hr of MCAO treatment followed by 24 hr reperfusion. The 2 hr MCAO was lethal. (**D**) RIP1-ASIC1a association in control (**C**) and MCAO hemisphere (M) following MCAO with durations as indicated (IP: ASIC1a; IB: RIP1). The levels of ASIC1a and RIP1 were assessed by IB. Note: total levels of RIP1 were reduced in the MCAO hemisphere compared to control, despite the increased association with ASIC1a. (**E**) Ischemia-induced RIP1 phosphorylation was abolished in *Asic1a⁻/⁻* brain. Upper, phosphorylation of RIP1 in ischemic brain of wild type (WT) mice, which was largely removed by 1 hr phosphatase treatment. Lower, *Asic1a* gene deletion prevented the ischemia-induced increase in RIP1 phosphorylation. (**F**) Schematic of a possible mechanism of acid-induced necroptosis. Upper, under normal physiological pH, the C-terminus of ASIC1a is protected by being buried inside or bound by an unknown protein; lower, acid stimulation exposes the CP-1 region of the ASIC1a C-terminus, allowing for association with and activation of RIP1, which in turn leads to necroptosis.

at least 2 hr (*Figure 5D*), the longest duration of MCAO tested (*Figure 5C*). The long time window of ASIC1a-RIP1 complex formation is consistent with the notion that acidosis occurs slowly and progressively in the ischemic brain and both acidosis and necroptosis contribute mainly to delayed ischemic brain injury (*Degterev et al., 2005*; *Pignataro et al., 2007*), a phase of neuronal damage with high clinical relevance because of the need for post-ischemic neuroprotection following stroke. Interestingly, the total protein level of RIP1 tended to decrease in the ischemic side of the brain (*Figure 5D*), similar to that observed in cultured cortical neurons subjected to acid treatment (*Figure 2F1,F2*). This lends additional support to the involvement of a similar regulatory mechanism between acid-induced neuronal death in vitro and ischemic brain damage in vivo. The phosphorylation of RIP1 is a critical event in brain ischemia and the blockade of RIP1 function with Nec-1 can significantly prevent ischemic brain injury (*Degterev et al., 2005*, *2008*). Consistent with previous reports (*Degterev et al., 2005*, *2008*), we observed increased RIP1 phosphorylation in ischemic brain from WT mice, which was successfully removed by treating the RIP1 immunoprecipitant for 1 hr with a bovine intestinal alkaline phosphatase (*Figure 5E*, upper). By contrast, the level of RIP1 phosphorylation in the ischemic hemisphere of the *Asic1a$^{-/-}$* mice remained unchanged as compared to the control hemisphere (*Figure 5E*, lower), demonstrating the critical role of ASIC1a in ischemia-induced RIP1 activation.

## Discussion

In the present study, we describe a novel form of neuronal necroptosis induced by extracellular acidosis and mediated by ASIC1a. Compared to conventional DR-dependent necroptosis, ASIC1a-dependent necroptosis did not require de novo synthesized ligands and extracellular protons may serve as the fast 'extrinsic death signal'. ASIC1a-dependent necroptosis does not require intrinsic ROS generation (*Figure 1B*, *Figure 1—figure supplement 3*), suggesting a different molecular mechanism from TNF-α-induced necroptosis. It has been shown previously that some forms of necroptosis are autophagy-dependent and do not rely on ROS (*Bonapace et al., 2010*). Notably, acidosis was reported to induce cell autophagy (*Wojtkowiak and Gillies, 2012*), suggesting a possible autophagy mechanism underlying ASIC1a-dependent necroptosis. Thus, further studies are needed to elucidate the detailed underlying mechanism(s) and clarify the similarities/differences between DR-dependent and ASIC1a-dependent necroptosis. Furthermore, as mentioned in the Introduction, deletion of the *Asic1a* gene strongly protects against ischemic neuronal death in a mouse model of focal ischemia (*Xiong et al., 2004*). A similar neuroprotective effect of Nec-1 was also observed in the same model (*Degterev et al., 2005*). Additionally, both ASIC1a and RIP1 were reported as potential therapeutic targets in traumatic brain injury (*You et al., 2008*; *Yin et al., 2013*). Importantly, we show here that RIP1 was recruited to ASIC1a in ischemic brains and the loss of ASIC1a prevented ischemia-induced RIP1 phosphorylation (*Figure 5A–E*), suggesting that ASIC1a-mediated RIP1 activation is a key step in ischemic brain injury. The inability to activate RIP1 probably explains the resistance of *Asic1a$^{-/-}$* mice to ischemic brain damage. These findings, altogether, strongly suggest ASIC1a is an important up-stream factor regulating RIP1 in vivo. Because both ASIC1a and RIP1 are widely expressed in nervous systems (*Degterev et al., 2005*; *Wemmie et al., 2006*) and tissue acidosis is a common feature of many neurological diseases, this novel DR-independent necroptosis probably contributes to neuronal injury in a broad range of neurological disorders.

We recently demonstrated the expression of ASIC1a in mitochondria (mtASIC1a) and found that it plays an important role in ROS-induced and mitochondrial permeability transition (MPT)-dependent neuronal death (*Wang et al., 2013*). The mtASIC1a functions quite differently from the plasma membrane ASIC1a. Although ASIC1a null neurons resisted $H_2O_2$-induced neuronal death, this effect was not reproduced with treatment with PcTX1, which cannot gain access to mtASIC1a localized in the inner mitochondrial membrane (*Wang et al., 2013*). However, PcTX1 effectively inhibited acid-induced neuronal death through blocking plasma membrane ASIC1a. Furthermore, whereas mtASIC1a mainly contributed to ROS-induced neuronal death, ROS production was not involved in acid-induced neuronal death (*Figure 1B*, *Figure 1—figure supplement 3*). Therefore, the death mechanism examined in the current study does not appear to involve mtASIC1a.

Here we show the indispensible role of ASIC1a channels in mediating neuronal necroptosis in response to extracellular acidification, but the ion-conducting function of the ASIC1a channels appears to not be essential for this process. Although it cannot be ruled out that ionic fluxes through the channel pore may play a modulatory role, our data suggest that RIP1 activation but not ionic

conduction per se is necessary for acid-induced necroptotic cell death (*Figure 5F*). Since ion channel activation reflects conformational changes induced by ligands and other gating factors, it is plausible that the acid-induced conformational change in ASIC1a channels exposes its proximal C-terminal region, represented by the CP-1 peptide (*Figure 4*), to trigger RIP1 phosphorylation (*Figure 5F*) and the consequent neuronal necroptosis in response to tissue acidosis. Ironically, steady-state desensitization of the ASIC1a channel by moderate pH decreases, although it suppressed subsequent activation of ASIC1a current, did not prevent acid-induced neuronal death (*Figure 3B*). This differed from the neuroprotective effect of PcTX1, which is believed to inhibit ASIC1a activation by causing steady-state channel desensitization at neutral and alkaline pH (*Chen et al., 2005*, *2006*). Presumably, extracellular protons cause at least two steps in conformational changes in ASIC1a: one that exposes the C-terminal RIP1 activation domain and the other that mediates channel gating. Only the latter is blocked by the steady-state desensitization induced by the moderated pH decrease, whereas both are inhibited by PcTX1, especially because of the presence of PcTX1 throughout the entire period of pH 6.0 treatment. This could not be the case for pH 6.8 pretreatment as the switch to the pH 6.0 solution necessarily eliminated the condition (pH 6.8) that caused steady-state desensitization in the first place and the C-terminal RIP1 activation domain eventually adopts the acid-induced conformation. Consistent with the two-step conformational changes, AMI was protective against neuronal death when applied before, but not during acidosis, which differed from its action in blocking channel conductance (*Figure 3—figure supplement 4B,C*).

Conduction-independent functions have been shown to contribute to various physiological and pathological processes for several other ion channels (*Kaczmarek, 2006*; *Levitan, 2006*). For example, the ether-à-go-go (EAG) K$^+$ channel modulates cell proliferation via a mechanism independent of K$^+$ flux conducted by the channel (*Hegle et al., 2006*) and certain Na$^+$ channel β subunits contribute to cell adhesion without requiring Na$^+$ permeability (*Kaczmarek, 2006*). Some channels are tightly associated with signaling molecules such as enzymes (*Kaczmarek, 2006*; *Levitan, 2006*). A typical example is the EAG channel, which regulates the activity of the mitogen-activated protein kinase (MAPK) pathway (*Hegle et al., 2006*). In another example, GluN2B-containing NMDA receptors are directly bound to phosphatase and tensin homolog deleted on chromosome ten (PTEN), cyclin-dependent kinase 5 (cdk5), and death-associated protein kinase 1 (DAPK1), to contribute crucially to excitotoxicity in ischemic brain (*Lai et al., 2014*). Others are multifunctional proteins containing clearly separate enzymatic and channel domains (*Kaczmarek, 2006*). For example, the melastatin-related transient receptor potential 7 (TRPM7) channel contains a kinase domain at its cytoplasmic C-terminus (*Runnels et al., 2001*). For ASIC1a channels, whether this occurs directly via auto-phosphorylation of RIP1 with its own kinase activity or indirectly by other associated kinase(s) requires further experimentation.

Although non-conducting functions have been reported for nearly every major class of ion channel (*Kaczmarek, 2006*), it remains unclear whether they represent exceptional cases in a few channel proteins or a general, but unrecognized, property of most ion channels. It is not uncommon for a protein to be multifunctional, but for an ion channel, the ion-conducting function probably attracts the most attention because of the well-developed methodologies and the rich information that can be acquired to characterize the channel. As a result, the non-conducting functions of ion channels have frequently been overlooked. This appears to be the case for ASIC1a. Ever since the protective role of *Asic1a* gene deletion on ischemic brain damage was reported, efforts have been made to understand the underlying mechanism(s) and nearly all reported studies focused on the ionic conduction, especially Ca$^{2+}$ influx, mediated by these channels (*Xiong et al., 2004*; *Yermolaieva et al., 2004*). However, as mentioned above and in accordance with the new evidence shown in the current study, the ionic conductance and Ca$^{2+}$ toxicity hypothesis is incompatible with the observed time dependence of severity of acid-induced neuronal death, at least under in vitro conditions. Interestingly, the acid-induced [Ca$^{2+}$]$_i$ rise and whole-cell currents in neurons, although highly correlated with ASIC1a expression, only occurred with fast focal application of the acidic solution (*Figure 3—figure supplement 2,3*), a condition that rarely happens during the development of tissue acidosis. Previously, partial protection was attained by reducing extracellular [Ca$^{2+}$] to 0.2 mM, leading to the conclusion that Ca$^{2+}$ influx was involved in acidic neuronal death (*Xiong et al., 2004*). However, the reduced extracellular [Ca$^{2+}$] could cause other complications that compromised necroptotic death. Therefore, we removed extracellular Ca$^{2+}$ completely and observed no neuroprotection in the present study. We further show that the ability of ASIC1a to mediate

acidotoxicity is independent of its ion-conducting function, but requires a C-terminal region of the channel protein. The sequence of CP-1 peptide that mimics acidosis in causing RIP1 phosphorylation and neuronal death represents a non-conserved region of ASIC C-termini (*Figure 4—figure supplement 1A*), raising the possibility that the conduction-independent necroptotic effect of ASIC1a channels arose later in evolution than the channel conducting function, which is universal for nearly all ASICs.

For neurons, a popular hypothesis is that protons co-released with neurotransmitters from acidic synaptic vesicles could activate postsynaptic ASIC1a, regulating physiological functions such as synaptic transmission, neuronal excitability, and learning/memory (*Wemmie et al., 2006*, *2013*). However, although *Asic1a*$^{-/-}$ mice exhibit abnormal synaptic activity and learning/memory deficits (*Wemmie et al., 2002*, *2006*; *Cho and Askwith, 2008*; *Urbano et al., 2014*), the non-substantial contribution of ASIC1a channels to synaptic events, such as excitatory postsynaptic potentials (EPSPs), is insufficient to account for the functions of these channels (*Alvarez de la Rosa et al., 2003*; *Cho and Askwith, 2008*; *Kreple et al., 2014*). It might be possible that a conduction-independent mechanism was also involved in the synaptic function of ASIC1a channels. A knock-in mouse model carrying the non-conducting HIF-ASIC1a mutant would be a useful tool to examine this possibility in future studies.

## Materials and methods

### Focal ischemia

The experimental protocols (ethics protocol number: 2014022) were approved by the Animal Care and Use Committee of Shanghai Jiao Tong University School of Medicine, Shanghai, China. A transient focal ischemia model was prepared as described previously (*Xiong et al., 2004*; *Duan et al., 2011*). Briefly, animals (male C57BL/6 mice, ~25 g) were anesthetized using 10% chloral hydrate with intubation and ventilation. Rectal and temporalis muscle temperature was maintained at 37 ± 0.5°C with a thermostatically controlled heating pad and lamp. A suture occlusion was made to the middle cerebral artery while cerebral blood flow (CBF) was monitored by transcranial laser Doppler. Animals whose blood flow did not reduce below 20% were excluded. For co-IP experiments, animals were killed and brains were removed immediately after MCAO of various durations of 0.5, 1, 1.5 and 2 hr. In other animals the suture was removed to allow reperfusion and the mice were euthanized 24 hr later. Brains were removed, sectioned coronally at 1 mm intervals, and stained with the vital dye 2,3,5-triphenyltetrazolium hydrochloride (TTC). Under the present experimental conditions, 2 hr MCAO was lethal to the animals.

### Primary culture of mouse cortical neurons

Postnatal day 1 C57BL/6 WT or *Asic1a*$^{-/-}$ mice (with a congenic C57BL/6 background) were anesthetized with halothane. Brains were removed rapidly and placed in ice-cold Ca$^{2+}$- and Mg$^{2+}$-free phosphate-buffered saline (PBS). Tissues were dissected and incubated with 0.05% trypsin-EDTA for 15 min at 37°C, followed by trituration with fire-polished glass pipettes, and plated in poly-D-lysine-coated 100 mm culture dishes (1 × 10$^7$ cells per dish) or 24-well plates (1.5 × 10$^6$ cells per well). Neurons were cultured with Neurobasal medium supplemented with B27 and maintained at 37°C in a humidified 5% CO$_2$ atmosphere incubator. Cultures were fed twice a week and used for all the assays 14–16 days after plating. Glial growth was suppressed by the addition of 5-fluoro-2-deoxyuridine (20 μg/ml; Sigma–Aldrich, St. Louis, MO) and uridine (20 μg/ml; Sigma–Aldrich, St. Louis, MO).

### shRNA of mouse RIP1

The virus-based RIP1 shRNA and negative control plasmids which had been successfully used in a previous study (*Zhang et al., 2009*) were kindly provided by Dr JH Han (School of Life Science, Xiamen University, Xiamen, China). Briefly, RIP1 shRNA was designed to target mouse RIP1 with the sequence GCATTGTCCTTTGGGCAAT, and its effectiveness was tested by Western blotting. shRNA targeting an irrelevant gene β-galactosidase was used as a negative control with the sequence TTGGATCCAA. The cultured mouse cortical neurons were infected with lentivirus for RIP1 shRNA or the negative control shRNA at DIV 7. Assays were performed 7 days after virus infection.

## Transfection of CHO cells and *Asic1a*$^{-/-}$ neurons

Cortical neurons from *Asic1a*$^{-/-}$ mice were cultured in no. 0 glass bottom dishes coated with poly-D-lysine for 5 days and a half volume of medium removed for later use before transfection. CHO cells were grown in 35 mm dishes, 24-well plates, or glass coverslips for 1 day. Transfection was carried out using HilyMax (Dojindo, Japan) according to the standard protocol. Briefly, neurons (in 1 ml of medium per culture) or CHO cells (in 2 ml of medium per culture) were transfected with 0.5–1 µg of the desired plasmid: EGFP-vector, EGFP-tagged or untagged WT-ASIC1a (human, EU078959.1), HIF-ASIC1a, RC-ASIC1a, ASIC1b (rat, EDL86977.1), ASIC2a (rat, NM_001034014.1), ASIC3 (rat, NM_173135.1), or TRPV1 (rat, NM_031982) and 2 µl HilyMax. For neurons, the transfection medium was replaced after 6 hr by a 1:1 mixture of the medium removed before the transfection and fresh culture medium. Cells were used at 48 hr after transfection. The GFP signal was used for the identification of transfected cells.

## Co-immunoprecipitation

Cultured mouse cortical neurons or brain tissues were collected and re-suspended in a lysis buffer [20 mM Tris-Cl, pH 7.4, 150 mM NaCl, 1% Triton X-100, 1 mM EDTA, 3 mM NaF, 1 mM β-glycerophosphate, 1 mM sodium orthovanadate, 2 mM *N*-ethylmaleimide and 10% glycerol, complete protease inhibitor set (Sigma–Aldrich, St. Louis, MO), and phosphatase inhibitor set (Roche, Switzerland)]. The re-suspended lysates were vortexed, incubated on ice for 40 min, and centrifuged at 13,000 × g for 15 min. The supernatant was incubated with 4 µg antibody overnight at 4°C. The following day, 20 µl protein G agarose beads were added to the sample and incubated for 2 hr at 4°C. Then, the beads were washed three times with the lysis buffer and the immunoprecipitants eluted with 2× loading buffer and subjected to Western blot analysis.

## Electrophysiological recordings

ASIC currents were recorded using whole-cell patch-clamp techniques at room temperature (22–25°C). For voltage-clamp recordings, the membrane voltage was held at −60 mV. The standard external solution (SS) contained: 150 mM NaCl, 5 mM KCl, 1 mM MgCl$_2$, 2 mM CaCl$_2$, and 10 mM glucose, buffered to various pH values with 10 mM HEPES. The osmolarity of all solutions was kept at 300–330 mOsm/l. In NMDG-replacement and Ca$^{2+}$-free experiments (*Figure 3D,E*), external solutions were adjusted accordingly. The patch pipette solution contained: 120 mM KCl, 30 mM NaCl, 1 mM MgCl$_2$, 0.5 mM CaCl$_2$, 5 mM EGTA, 4 mM Mg-ATP, and 10 mM HEPES, pH 7.4. All drugs for electrophysiological experiments were purchased from Sigma–Aldrich (St. Louis, MO). A Y-tube apparatus was used for drug administration in electrophysiological and Ca$^{2+}$-imaging (see below) experiments. The inner diameter of the open end of the Y-tube was ~100 µm. The flow rate was adjusted by changing the height of the solution reservoir. The tip of the Y-tube was placed ~500 µm away from the cell body in order to ensure complete exposure to the perfusion solution by the cells being recorded without distorting the cell shape due to solution flush.

## Ca$^{2+}$ imaging

Cultured mouse cortical neurons grown on no. 0 glass bottom dishes were washed three times with SS (pH 7.4) and incubated with 1 µM Fura-2 AM for 30 min at 37°C, followed by washing three times with SS. The dish was mounted on the stage of an inverted fluorescence microscope (Nikon Eclipse TI, Japan) and neurons were observed with a 20× objective lens. Fura-2 fluorescence images were acquired with alternating excitation wavelengths of 340 and 380 nm and an emission wavelength of 510 nm at 0.5 Hz while cells were continuously perfused with a Y-tube placed approximately 500 µm from the cells at a rate of ~50 µl/min (fast perfusion) or ~15 µl/min (slow perfusion). The acidic (pH 6.0) solution was applied through the Y-tube at the same rate. To block secondary activation of glutamate receptors and voltage-gated Na$^+$ and Ca$^{2+}$ channels, 20 µM AP5, 20 µM CNQX, 1 µM TTX, and 5 µM nimodipine were included in the perfusate. PcTX1 (50 nM) was included to inhibit ASIC1a. The Fura-2 ratio (340/380) was used to represent [Ca$^{2+}$]$_i$ changes.

 CHO cells grown on glass coverslips transfected with mCherry-ASIC1a were washed twice with SS (pH 7.4) and then incubated with 2 µM Fluo4-AM (Dojindo, Japan) in the presence of 0.02% Pluronic F-127 at 22°C for 60 min, followed by washing twice with SS (pH 7.4). The coverslip was transferred to a perfusion chamber, which was mounted on the stage of a Nikon Eclipse TI (Japan). Fluo4

fluorescence images were taken at 0.3 Hz with excitation and emission wavelengths of 488 and 520 nm, respectively. Solution changes were achieved with the use of a Y-tube at ~50 µl/min.

## Proton imaging

Cultured mouse cortical neurons grown on glass coverslips were incubated with 5 µM BCECF-AM at 37°C for 30 min and then washed twice with SS (pH 7.4). The coverslip was transferred to a perfusion chamber, which was mounted on the stage of a Nikon Eclipse TI (Japan). BCECF fluorescence images were taken with alternating excitation wavelengths of 490 and 440 nm and an emission wavelength of 535 nm at 0.1 Hz. Solution changes were achieved with the use of a Y-tube at ~50 µl/min.

## Death assay

Acid-induced neuronal death was achieved as described previously (*Xiong et al., 2004*). First, cells were washed three times with the treatment solution (150 mM NaCl, 5 mM KCl, 1 mM MgCl$_2$, 2 mM CaCl$_2$, and 10 mM glucose, buffered to the desired pH value with 10 mM HEPES) within 5 min at room temperature (22–25°C), and then incubated at 37°C for different time periods depending on experimental purposes. At the end of treatment, the solution was replaced with the normal pH culture medium and the culture resumed at 37°C for 24 hr. Cell viability was assessed by propidium iodide (PI) staining, lactate dehydrogenase (LDH) measurement, and the Cell Titer Blue (CTB) assay. Briefly, cells were stained with 10 µg/ml PI for 10 min at room temperature and then examined by fluorescence microscopy. NeuN-staining or DIC was used to distinguish neurons from glia. For the LDH assay, neurons were washed three times with the external solution and randomly divided into treatment groups. Neurons were washed and incubated in the normal culture medium at 37°C for 24 hr. The LDH level in the culture medium, indicative of cell death, was measured using the LDH assay kit (Roche Molecular Biochemicals, Switzerland). An aliquot of the medium (100 µl) was transferred from the culture wells to the wells of a 96-well plate and mixed with 100 µl of the reaction solution provided by the kit. Optical density was measured at 492 nm 45 min later using the SpectraMax Paradigm Multimode Microplate Reader (Molecular Devices, Sunnyvale, CA). Background absorbance at 620 nm was subtracted. The maximal releasable LDH in each well was then obtained by a 15 min incubation with 1% Triton X-100 at the end of each experiment. For the CTB assay, neurons were cultured in the wells of 24-well plates. The amount of culture medium was adjusted to the same in each well (0.5 ml) and pH 6.0 treatment was performed in the absence and presence of different drugs. After a return to normal culture for 24 hr, 0.1 ml CTB solution (Promega, Madison, WI) was added to each well and the plate incubated for 2 hr at 37°C. The fluorescence intensities (excitation, 560 nm; emission, 590 nm), indicative of the amounts of viable cells, were measured using the SpectraMax Microplate Reader. All death assays were performed with four to eight repeats each time.

## In vitro RIP1 auto-phosphorylation assay

Mouse cortices were collected and resuspended in a lysis buffer [20 mM Tris-Cl, pH 7.4, 150 mM NaCl, 1% Triton X-100, 5 mM EDTA, 3 mM NaF, 1 mM sodium orthovanadate, 10% glycerol, and complete protease inhibitor set (Sigma–Aldrich, St. Louis, MO)]. The lysates were vortexed for 20 s and then incubated on ice for 40 min and centrifuged at 13,000 rpm for 15 min. The supernatant was incubated with 4 µg antibody against RIP1 (BD Biosciences, San Jose, CA, 610458) overnight at 4°C. The next day, 20 µl protein G agarose beads were added to the sample and incubation continued for 2 hr at 4°C. Then, the beads were washed three times with the lysis buffer before being resuspended in 20 µl of the kinase reaction buffer (20 mM HEPES, pH 7.3, 5 mM MgCl$_2$, and 5 mM MnCl$_2$). This was followed by the addition of 20 µM of cold ATP to initiate the kinase reaction, which lasted for 1 hr at 30°C and was terminated by the addition of 20 µl of 2× SDS loading buffer. The samples were vortexed, centrifuged for 1 min at 13,000 rpm, heated to 95–100°C for 5 min, and then cooled on ice for 1 min. After another centrifugation for 5 min at 13,000 rpm, the supernatants were subjected to Western blot analysis using the anti-phospho-S/T antibody (Cell Signaling, Danvers, MA, phospho-PKA substrate, clone100 G7E) to detect phosphorylated RIP1.

## Dephosphorylation assay

Immunoprecipitation of RIP1 was performed as described above. The immunoprecipitants were washed three times with a dephosphorylation buffer (DB; 100 mM NaCl, 50 mM Tri-HCl, 10 mM MgCl$_2$, 1 mM dithiothreitol, pH 7.9, and Sigma complete protease inhibitor set), then incubated in DB

with 10 units of bovine intestinal alkaline phosphatase (Sigma–Aldrich, St. Louis, MO, P0114) at 37°C for 1 hr before Western blot analysis for phosphorylated RIP1 was performed as described above.

## $^{32}$P-labeling assay

Cultured mouse cortical neurons were washed three times with pre-warmed SS (pH 7.4 or pH 6.0). Then, 2 ml pre-warmed SS (pH 7.4 or pH 6.0) containing 1 mCi/ml $^{32}$P was added and cells were incubated for 30 min at 37°C. After incubation, the labeling medium was removed and the dish was washed with cold SS three times. Cells were then lysed in a lysis buffer [20 mM Tris-Cl, pH 7.4, 150 mM NaCl, 1% Triton X-100, 1 mM EDTA, 3 mM NaF, 1 mM β-glycerophosphate, 1 mM sodium orthovanadate, 2 mM *N*-ethylmaleimide and 10% glycerol, complete protease inhibitor set (Sigma–Aldrich, St. Louis, MO), and phosphatase inhibitor set (Roche, Switzerland)], vortexed, incubated on ice for 40 min, and centrifuged at 13,000 × g for 15 min. The supernatant was incubated with 4 μg anti-RIP1 antibody overnight at 4°C. The next day, 20 μl protein G agarose beads were added to the sample, which was then incubated for 2 hr at 4°C. The beads were washed three times with the lysis buffer, the immunoprecipitants eluted with 2× loading buffer and were subjected to Western blot analysis. The gel was exposed to a phosphorimaging screen overnight to detect phosphorylated RIP1. The screen was scanned using the STORM imaging system.

## Statistical analysis

Statistical comparisons were performed using unpaired or paired Student's $t$ tests (for data with non-normal distribution, the Kolmogorov–Smirnov test was used) where values of $p < 0.05$ are considered significant.

## Acknowledgements

We thank Professor MJ Welsh (Howard Hughes Medical Institute, University of Iowa, Iowa City, IA) for providing *Asic1a* knockout mice, Dr JH Han (Xiamen University, Xiamen, China) for kindly providing virus-based RIP1 shRNA and negative control plasmids, and Miss Q Jiang for technical support. This study was supported by grants from the National Basic Research Program of China and the National Natural Science Foundation of China (nos. 2014CB910300, 313111222, 91413122, 91213306, 91132303, and 31230028), the US National Institutes of Health (R01 GM092759, U54NS083932), and the American Heart Association (15GRNT23040032). Yi-Zhi Wang was a postdoctoral fellow supported by funding from the China Postdoctoral Science Foundation.

# Additional information

### Funding

| Funder | Grant reference | Author |
|---|---|---|
| National Basic Research Program of China | 973 Program | Tian-Le Xu |
| National Basic Research Program of China | 2014CB910300 | Tian-Le Xu |
| National Natural Science Foundation of China | 313111222 | Tian-Le Xu |
| National Natural Science Foundation of China | 91413122 | Tian-Le Xu |
| National Natural Science Foundation of China | 91213306 | Tian-Le Xu |
| National Natural Science Foundation of China | 91132303 | Tian-Le Xu |
| National Natural Science Foundation of China | 31230028 | Tian-Le Xu |
| National Institutes of Health | U54NS083932 | Michael X Zhu |

| Funder | Grant reference | Author |
| --- | --- | --- |
| National Institutes of Health | U54NS083932 | Zhi-Gang Xiong |
| China Postdoctoral Science Foundation | 2012M511106 | Yi-Zhi Wang |
| American Heart Association | 15GRNT23040032 | Michael X Zhu |

The funders had no role in study design, data collection and interpretation, or the decision to submit the work for publication.

### Author contributions

Y-ZW, Conception and design, Acquisition of data, Analysis and interpretation of data, Drafting or revising the article, Contributed unpublished essential data or reagents; J-JW, YH, FL, W-ZZ, YL, Acquisition of data, Contributed unpublished essential data or reagents; Z-GX, Drafting or revising the article, Contributed unpublished essential data or reagents; MXZ, T-LX, Conception and design, Analysis and interpretation of data, Drafting or revising the article

### Ethics

Animal experimentation: The experimental protocols (ethics protocol number: 2014022) were approved by the Animal Care and Use Committee of Shanghai Jiao Tong University School of Medicine, Shanghai, China.

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
