## [Decision Letter]

Thank you for sending your work entitled “Acid sensing and signaling in the ischemic brain: Acidotoxicity revisited” for consideration at *eLife*. Your article has been evaluated by a Senior Editor, a Reviewing Editor, and three reviewers.

The Reviewing editor and the reviewers discussed their comments before we reached this decision, and the Reviewing editor has assembled the following comments to help you prepare a revised submission. All reviewers agreed that the manuscript presents an intriguing set of experiments that have been carefully combined to support a highly innovative hypothesis: that ASIC-mediated neuronal death involves a novel form of necroptosis induced by phosphorylation of the kinase RIP1 via an acid and ASIC1-dependent mechanism that does not rely on ASIC ion conduction. However, they added the following cautions: some of the highly innovative ideas presented within this manuscript are based on negative data and experiments that fail to replicate results previously reported by the authors themselves and others. Therefore, the methodology must be clear and all important controls should be reported. Many of these controls are missing and interpretations that would support a less exciting explanation of the results have not been ruled-out. Thus, the major conclusion that the ASIC-dependent neuronal cell death is independent of Ca^2+^ change and channel conductance, although very appealing, appears to be premature and requires further support.

Key issues noted by the reviewers (and detailed below under “Specific comments”) include whether HIF-ASICa is really non-conducting and the fact that permeant ions (protons) remain even in the ion substitution experiments. Of major interest is some resolution of the discrepancy with the previous study of [50]. The reviewers agreed that, to be definitive, it will be necessary to characterize the permeability of the mutant, add the calcium imaging data as requested below, repeat the CHO studies in neurons, and utilize additional methodology to assess cell death (since ASICs affect mitochondrial function and the prominent assay used doesn't allow definitive conclusions).

Specific comments:

1) The exact methodology of acid-mediated death (solutions/timing/quantification, etc) is not described or referenced. The lack of this information makes is impossible to properly review the manuscript and place the results within the context of previously published work that have reported opposite results.

2) The bulk of the work in this manuscript uses the CTB assay, which measures reduction of the redox dye (resazurin) into a fluorescent end product (resorufin). This is dependent on the number of viable cells as well as mitochondrial metabolism. The authors previously reported that ASIC1a affects mitochondrial activity and cell death (Wang et al., Cell death and Differentiation, 2013). Although no reference to this paper is made within the current manuscript, it seems likely that the interventions could alter mitochondrial localization of ASICs or mitochondrial function to affect the results of the CTB viability assay. A secondary method should be used to rule out non-specific effects due to changes in ASIC localization to the mitochondria (which is not being measured) or pH-dependent changes in mitochondrial activity independent of ASICs.

3) The authors show that steady state desensitization of ASIC1a does not prevent acidosis-induced neuronal death (Figure 3) in opposition to the results reported by Sherwood et al., J. Neurosci 2009. In Figure 3, control data should be presented in which cells were *not* exposed to pH 6.8 for 2 minutes to allow determination if such 6.8 incubation attenuated acid-mediated death at all. Further, perfusion systems for patch clamp (Figure 3) are often much faster and more accurate than the solution exchange within a 100 mm culture or 24 well dish (Figure C and D). Thus, the authors should also report how they directly measured the pH of the solution within the viability assay to ensure that their methodology allowed pH 6.8 to be obtained in the tissue culture wells prior to pH 6.0 incubation.

4) Figure 3 shows that removal of calcium or in fact all major ions from the treatment solution does not impact acidosis-induced changes in cell viability as measured with the CTB assay. This is a dramatic reversal of the seminal observation that ASIC1a-mediated neuronal death is dependent on extracellular calcium (Xiong et al., Cell, 2004 showed; Figure 5) and should be well supported. A control should be included which illustrates that pH 6.0-evoked decrease in viability using the CTB assay under these specialized conditions is still dependent on ASIC1a (i.e. PcTx1-mediated inhibition). Ca^2+^ imaging experiments are needed to establish that the low pH-induced cell death is not dependent on increases in [Ca^2+^]_i_. Finally, Ca^2+^-free medium also induces neuronal depolarization, which could complicate the interpretation of the data in Figure 3.

5) Removal of sodium, potassium, and calcium has not removed all the permeant ions. ASIC1a is also permeant to protons (Waldmann et al. Nature 386, 1997; Chen and Grunder, J. Physiology 579, 2007). Given this fact, it is difficult to see how ion conduction can be completely ruled out.

6) The authors measure acidosis-induced neuronal death in stable CHO cell lines (Figure 2) and interchange these data with those obtained in neurons. The methodology for the viability in CHO cells is unclear and interpretations difficult to solidify. Use of such methodology requires that the percentage of cells expressing ASIC1a be equivalent between groups. Further, CHO cells are actively dividing and acidosis-induced neuronal death is measured 24 hours after exposure to pH 6.0. Use of the CTB viability assay measures the number of viable cells. Thus, any difference could be due to pH 6.0-evoked changes in cell death or a cell division. Controls should be presented that acid-mediated death utilizes equivalent mechanisms in neurons and CHO cells (i.e. RIP1 dependent, PcTx1 susceptible, etc). Alternatively, these experiments could be done in transfected neurons.

7) The most compelling data to support the idea that ASICs mediate neuronal death independent of ion conduction comes from the use of HIF-ASIC1a in CHO cells. The authors state that “while both were expressed normally on the plasma membrane (Figure 3—figure supplement 2), HIF-ASIC1a (32HIF34 mutated to 32AAA34) was non-conducting due to pore dysfunction (29).” Yet, [29] did not study this mutant and HIF-ASIC1a has not been previously characterized or linked to the pore. Pfister et al. did study these residues, but felt that their data “did not allow them to conclude” that they impact the pore or channel gating (see discussion of [29]). In fact, recent data with ENAC (Kucher et al., Biophys J 100(8) 1930–1939, 2011) suggest that channels with mutations in these residues might, in fact, be conductive in a voltage dependent manner and only appear non-functional in traditional voltage-clamp conditions. It is also unclear whether the HIF1 mutation alters proton permeation or whether HIF-ASIC1a localized to mitochondria.

8) The authors find that co-administration of amiloride (unknown concentration) with pH 6.0 eliminates acid-gated currents, but does not affect acidosis-induced changes in cell viability (Figure 3—figure supplement 1. This seems in direct opposition to the results of Xiong et al., Cell 2004, (Figures 4 and 5). Similarly, the current manuscript suggested that pretreatment with amiloride for 1 hour did prevent acid-dependent changes in cell viability. Xiong et al., previously reported that incubating neurons for 10 min prior and during acidosis with amiloride prevented toxicity and amiloride incubation for extended periods of time is toxic (Xiong et al., Cell 2004). Additional information on the methodology as well as some inclusion of a discussion of the difference in these results should be provided.

9) PcTx1 inhibits ASIC currents by pushing the channel into the steady state desensitized state. If steady state desensitization is still toxic, then PcTx1 must be attenuating acid induced neuronal death by preventing the “non-conductive” conformational transition which activates RIP1-mediated death in manner similar to that suggested for 1 hour amiloride. This should be addressed within the Discussion.

10) The authors show that the CP1 region of the C-terminus is toxic to neurons in a manner that is independent of acidic pH and partially reversed by Nec-1. Given the non-specificity of Nec-1 and the basic nature of the included ASIC1-C-terminal fragment, it is important to further explore this toxicity. Is the effect dependent on RIP1 or ASIC1a? The importance of the TAT peptide should also be assessed as a control.

11) The most direct evidence supporting the idea that the low-pH induced ASIC-dependent neuron death does not require the channel's ionic conduction is from Figure 3 where the authors show that a non-conducting ASIC (HIF-ASIC1a) is as efficient as the wild-type in mediating pH6.0-induced cell death. However, this experiment was done in CHO fibroblast cells. The cell death is only ∼30% (∼20% higher than in the “blank” control), way below the ∼80% observed in neurons (Figure 1). At minimum, the authors should test the two constructs in the ASIC knockout neurons and test whether they have similar ability in restoring the pH-induced cell death.

12) The authors rely heavily on pharmacology. There is a general lack of discussion of why drugs were used at certain concentrations and applied for particular durations; how specific they are at those concentrations; and why the situation is thought to be the same between the various cells/tissues studied.

13) Along the similar lines, the authors use the caspase inhibitor, zVAD-fmk, as a major part of their argument that cells are dying by necroptosis and not apoptosis. It would be useful to have a positive control for this agent (e.g., by showing that it blocks the effect of staurosporine).

14) What pH is induced in the brain during MCAO? The authors should estimate this value and, if it is very different from 6.0 (the value tested in in vitro experiments), they should present additional evidence that the mechanism they worked out in vitro is operative in vivo.

15) There is insufficient demonstration of the necroptotic phenotype at the level of morphology. The authors provide EM images of cells but these are few and not quantified.

[Editors' note: further revisions were requested prior to acceptance, as described below.]

Thank you for resubmitting your work entitled “Tissue acidosis induces neuronal necroptosis via ASIC1a channel independent of its ionic conduction” for further consideration at *eLife*. Your revised article has been favorably evaluated by a Senior Editor, a Reviewing Editor, and two reviewers. The manuscript has been greatly improved but there are some remaining issues that need to be addressed before acceptance, as outlined below:

The reviewers acknowledge the strength of the ionic substitution (NMDG) and pore mutant experiments at making a role for Ca influx through ASICs appear highly unlikely for necroptosis. Nevertheless, they note that the novelty of complete Ca–independence is substantial and also deviates from previous work from your group, and therefore requires evidence not only that increased Ca levels are unnecessary for necroptosis but also that Ca levels do not actually increase (e.g., through another mechanism; if the mechanism is distinct, of course the case can still be made that Ca may not enter through the ASIC pore). Reviewer 1 suggests an imaging experiment in neurons (rather than just CHO cells); Reviewer 2, with the same underlying concerns, requests a discussion of possible bases for the difference between this and previous work.

Reviewer #1 (General assessment and major comments):

The authors have responded to the previous reviews with additional experiments and the manuscript has been strengthened. It will be an important manuscript for the field. The one significant omission that remains is calcium imaging in neurons. The authors should report ASIC1a-dependent calcium signal in neurons treated with Nec-1 and in neurons transfected with the different ASIC1a constructs used in the death assays. The authors now show that acid-induced calcium signaling is absent in transfected CHO cells (this has been previously reported by Samways et al., as stated by the authors) and that acid-induced cell death occurs in the absence of extracellular calcium in neurons. Yet, as the authors eloquently discuss, calcium permeability and acid-induced calcium fluctuations have been the focus of the field for years. The irrelevance of calcium is such a significant reversal of previous data published by the authors as well as almost all reviews of ASIC1a-dependent processes. This assertion should be backed by all commonly available methodologies. As the authors have previously reported imaging of ASIC1a-induced calcium signals in neurons (Gao et al., Neuron, 2005), these measures of calcium are conspicuously absent. Given the importance of these results for the field, they should also be included.

Reviewer #2 (General assessment and major comments):

The authors have done a good job addressing most of the questions raised by the reviewers. The data looks interesting and it reveals a novel mechanism for low pH-induced neuronal death.

Reviewer #2 (Minor comments):

One thing that still puzzles me, and perhaps other readers, is the nearly total lack of Ca^2+^ influx-dependence of low pH-induced cell death as observed in this paper; yet the [50] paper demonstrated clear Ca^2+^-dependence. The experimental conditions in the two studies appear to be similar. The authors did not address this apparent discrepancy as asked by the reviewers. They should at least include some discussion to address this.

[Editors' note: further revisions were requested prior to acceptance, as described below.]

Thank you for resubmitting your work entitled “Tissue acidosis induces neuronal necroptosis via ASIC1a channel independent of its ionic conduction” for further consideration at *eLife*. Your revised article has been favorably evaluated by a Senior Editor, a Reviewing Editor, and one reviewer. The manuscript has been improved but there are some remaining issues that need to be addressed before acceptance, as outlined below:

The new experiments on the response of neurons to changes in pH raise the interesting possibility that the rate of solution application is the variable that accounts for the differences between the present and previous work. However, as pointed out by the reviewer, the information given in the manuscript suggests that the pH change in the slow-flow condition was so slow that it raises the concern that the cells were never exposed to the solution at all, which would explain the lack of Ca signal detected. The editors recognize that this perception may be a misunderstanding, owing to an incomplete description of the Y-tube apparatus and/or how it was verified that the solution reached the cell. If so, please edit the manuscript to make the methods clearer, so that this concern can be alleviated. If, however, no verification was done, it may be necessary either to provide alternative evidence to justify the claim that flow rate is the key variable determining the Ca signals, or to remove that conclusion and relegate the hypothesis of flow rate to the Discussion.

Reviewer #2:

The authors addressed the concern of the reviewers by imaging low pH-induced Ca^2+^ changes in cortical neurons. They conclude that low pH perfusion leads to Ca^2+^ influx only when at high perfusion speed (50 µl/min) but not at low speed (15 µl/min). The authors further imply that the low speed, which induces no detectable Ca^2+^ increase, may be closer to the pathophysiological conditions. While I was generally satisfied with the previous version and did not specifically ask for additional measurements, I found the data added in this revision is totally unconvincing. The perfusion was done with focal perfusion using a Y-tube (opening size not specified) placed ∼500 µm away from the neuron. The concentration of the perfusion solution (i.e. the pH change) a neuron experiences is determined by the opening and the shape of the tube, and the angle at which the tube is aligned against the neuron. A simple explanation for the lack of Ca^2+^ change during slow perfusion could be that the neuron never experienced a significant pH change, and this had little to do with the speed of the pH change. After all, a15 µl/min perfusion against a dish (1 ml volume?) is very slow and there might be a large pH gradient between the tube opening and the cell body (even for a 500 µm distance). It's not clear to me why the authors did not use whole-dish perfusion (as done for the CHO cells) where the time it takes to change the whole bath can be more accurately measured/estimated. In summary, it appears to me that the conclusion that Ca^2+^ change is speed-dependent is meaningless, unless both the degree and the speed of pH change of bath surrounding the neuron are known.

---

## [Author Response]

*[…] Key issues noted by the reviewers (and detailed below under “Specific comments”) include whether HIF-ASICa is really non-conducting and the fact that permeant ions (protons) remain even in the ion substitution experiments. Of major interest is some resolution of the discrepancy with the previous study of*
[50]*. The reviewers agreed that, to be definitive, it will be necessary to characterize the permeability of the mutant, add the calcium imaging data as requested below, repeat the CHO studies in neurons, and utilize additional methodology to assess cell death (since ASICs affect mitochondrial function and the prominent assay used doesn't allow definitive conclusions).*

We thank the reviewers for the positive comments on our present work. We agree that the suggested experiments are critical to support our conclusion. We have performed all the experiments as suggested and obtained results that provide additional support to the non-conducting role of ASIC1a in acid-induced neuronal necroptosis (please see new Figure 1; Figure 1—figure supplement 1; Figure 1—figure supplement 2; Figure 2; Figure 2—figure supplement 1; Figure 3; Figure 3—figure supplement 1; Figure 3—figure supplement 2; Figure 3—figure supplement 4; Figure 3—figure supplement 5; Figure 3—figure supplement 6; Figure 4—figure supplement 1).

Specific comments:

1) The exact methodology of acid-mediated death (solutions/timing/quantification, etc) is not described or referenced. The lack of this information makes is impossible to properly review the manuscript and place the results within the context of previously published work that have reported opposite results.

We have described the exact methodology of acid-mediated death in the Materials and methods section and used diagrams to show experimental schemes (please see new Figure 1—figure supplement 1, Figure 3—figure supplement 1). Please also see the addition of a key reference (50) for the methodology used.

2) The bulk of the work in this manuscript uses the CTB assay, which measures reduction of the redox dye (resazurin) into a fluorescent end product (resorufin). This is dependent on the number of viable cells as well as mitochondrial metabolism. The authors previously reported that ASIC1a affects mitochondrial activity and cell death (Wang et al., Cell death and Differentiation, 2013). Although no reference to this paper is made within the current manuscript, it seems likely that the interventions could alter mitochondrial localization of ASICs or mitochondrial function to affect the results of the CTB viability assay. A secondary method should be used to rule out non-specific effects due to changes in ASIC localization to the mitochondria (which is not being measured) or pH-dependent changes in mitochondrial activity independent of ASICs.

We have now cited the paper (please see Discussion, second paragraph). The plasma membrane ASIC1a works as a 'death initiator', which directly responds to extracellular acidosis to initiate neuronal necroptosis, whereas the mitochondrial ASIC1a (mtASIC1a) contributes to the 'execution of death', which mainly mediates H_2_O_2_-induced neuronal death. Therefore, the specific inhibitor of ASIC1a, PcTX1, inhibited extracellular acidosis-induced neuronal necroptosis (please see new Figure 1, Figure 1—figure supplement 2), but not H_2_O_2_-induced neuronal death because of the inaccessibility of extracellularly applied PcTX1 to mtASIC1a located in mitochondrial inner membrane (please see Figure 1 in the paper by [42]). Nevertheless, to rule out non-specific effects due to potential changes in ASIC localization to the mitochondria acidic neuronal death was also assayed by measuring lactate dehydrogenase (LDH) release and counting PI-positive neurons (please see new Figure 1; Figure 1—figure supplement 2; Figure 3; Figure 3—figure supplement 1; Figure 3; and Figure 3—figure supplement 5). These two methods are independent of mitochondrial activity. The results are consistent with those acquired via CTB assays (please see new Figure 1), indicating that the CTB assay is a reliable method to examine acid-induced cell death under our experimental conditions. So far, we have no evidence supporting that mtASIC1a directly contributes to acid-induced neuronal necroptosis.

*3) The authors show that steady state desensitization of ASIC1a does not prevent acidosis-induced neuronal death (*Figure 3*) in opposition to the results reported by Sherwood et al., J. Neurosci 2009. In*
Figure 3*, control data should be presented in which cells were* not *exposed to pH 6.8 for 2 minutes to allow determination if such 6.8 incubation attenuated acid-mediated death at all. Further, perfusion systems for patch clamp (*Figure 3*) are often much faster and more accurate than the solution exchange within a 100 mm culture or 24 well dish (Figure C and D). Thus, the authors should also report how they directly measured the pH of the solution within the viability assay to ensure that their methodology allowed pH 6.8 to be obtained in the tissue culture wells prior to pH 6.0 incubation.*

We apologize for not making the experimental conditions clear for Figure 3. In Figure 3, neurons were treated with pH 6.0, *but not* pH 6.8 solutions, for the different periods of time as indicated. There was no pretreatment for any of the conditions shown. In our experimental system, it is impossible to replace the solution in 24-well dishes within 2 min because neurons needed to be washed for three times to ensure complete solution exchange. The shortest time we could reach was 10 min. Inspired by the reviewers’ question, we pre-treated the neurons with the pH 6.8 solution for 10 min and then exposed them to the pH 6.0 solution for 1 hr. We found no significant protective effect by the pretreatment with pH 6.8 solution (please see new Figure 3, Figure 3—figure supplement 1), suggesting that steady-state desensitization of the ASIC1a channel does not protect neurons from acid-induced death under our experimental conditions. To ensure accurate pH treatment, we rapidly washed neurons three times with the treatment solution at the beginning of the incubation and measured the pH value of the final solution by a pH meter immediately following the treatment.

*4)*
Figure 3
*shows that removal of calcium or in fact all major ions from the treatment solution does not impact acidosis-induced changes in cell viability as measured with the CTB assay. This is a dramatic reversal of the seminal observation that ASIC1a-mediated neuronal death is dependent on extracellular calcium (Xiong et al., Cell 2004 showed;*
Figure 5*) and should be well supported. A control should be included which illustrates that pH 6.0-evoked decrease in viability using the CTB assay under these specialized conditions is still dependent on ASIC1a (i.e. PcTx1-mediated inhibition). Ca*^*2+*^
*imaging experiments are needed to establish that the low pH-induced cell death is not dependent on increases in [Ca*^*2+*^*]*_*i*_*. Finally, Ca*^*2+*^*-free medium also induces neuronal depolarization, which could complicate the interpretation of the data in*
Figure 3*.*

We have performed the suggested experiments. In the absence of Ca^2+^, we found acid-induced neuronal death was significantly inhibited by PcTX1 and 7-Cl-O-Nec-1 (Nec-1s, please see new Figure 3—figure supplement 2), suggesting the involvement of the same ASIC1a- and RIP1-dependent death mechanism as in the normal Ca^2+^-containing solution. Furthermore, we observed that both HIF-ASIC1a, a non-conducting mutant, and ASIC1b, a Ca^2+^-impermeable variant, mediated acid-induced cell death (please see new Figure 3, Figure 3—figure supplement 6). These results also support that Ca^2+^ influx is not essential for ASIC1a-mediated cell death. Furthermore, we have measured intracellular Ca^2+^ changes in CHO cells expressing ASIC1a and found that the pH 6.0 solution elicited no detectable intracellular Ca^2+^ rise (please see new Figure 3—figure supplement 2). We agree with the reviewers that the Ca^2+^-free medium could induce neuronal depolarization, which complicates the data interpretation. That is why the rescue by PcTX1 and Nec-1 becomes important and the results on cell death mediated by HIF-ASIC1a and ASIC1b should also be taken into consideration to support the overall conclusion.

5) Removal of sodium, potassium, and calcium has not removed all the permeant ions. ASIC1a is also permeant to protons (Waldmann et al. Nature 386, 1997; Chen and Grunder, J. Physiology 579, 2007). Given this fact, it is difficult to see how ion conduction can be completely ruled out.

The proton-permeability of ASIC1a was demonstrated using the *Xenopus* oocyte expression system (Waldmann et al., 1997; Chen and Grunder, 2007). We have tried hard to measure proton currents mediated by ASIC1a in mammalian cells. However, in neither primary cultures of mouse cortical neurons nor CHO cells transfected with ASIC1a could we detect any proton-current (please see new Figure 3.). Based on the results of Chen and Grunder (Chen and Grunder, 2007), the fractional proton current of ASIC1a could be as high as 1-5% of the total current induced by acid (pH 6.0). In CHO cells, if the proton current existed, it would reach ∼100 – 500 pA (assuming a maximal acid-induced current of ∼10 nA) in the NMDG external solution, which would be easily detectable. However, we did not detect such currents, suggesting that most likely, ASIC1a channels do not conduct significant proton current in mammalian cells. Therefore, it is uncertain whether the proton permeability of ASIC1a is a universal phenomenon.

*6) The authors measure acidosis-induced neuronal death in stable CHO cell lines (*Figure 2*) and interchange these data with those obtained in neurons. The methodology for the viability in CHO cells is unclear and interpretations difficult to solidify. Use of such methodology requires that the percentage of cells expressing ASIC1a be equivalent between groups. Further, CHO cells are actively dividing and acidosis-induced neuronal death is measured 24 hours after exposure to pH 6.0. Use of the CTB viability assay measures the number of viable cells. Thus, any difference could be due to pH 6.0-evoked changes in cell death or a cell division. Controls should be presented that acid-mediated death utilizes equivalent mechanisms in neurons and CHO cells (i.e. RIP1 dependent, PcTx1 susceptible, etc). Alternatively, these experiments could be done in transfected neurons.*

We have repeated the key experiments using ASIC1a knockout neurons transfected with WT and mutant ASIC1a (please see new Figure 3). The results support the conclusion made from studying transfected CHO cells. In addition, we also examined acid-induced RIP1-ASIC1a association in transfected CHO cells. As in neurons, acid treatment recruited RIP1 to ASIC1a in CHO cells that expressed ASIC1a (please see new Figure 3—figure supplement 6).

*7) The most compelling data to support the idea that ASICs mediate neuronal death independent of ion conduction comes from the use of HIF-ASIC1a in CHO cells. The authors state that “while both were expressed normally on the plasma membrane (*Figure 3—figure supplement 2*), HIF-ASIC1a (32HIF34 mutated to 32AAA34) was non-conducting due to pore dysfunction (*[29]*).” Yet,*
[29]
*did not study this mutant and HIF-ASIC1a has not been previously characterized or linked to the pore. Pfister et al. did study these residues, but felt that their data “did not allow them to conclude” that they impact the pore or channel gating (see discussion of*
[29]*). In fact, recent data with ENAC (Kucher et al., Biophys J 100(8) 1930–1939, 2011) suggest that channels with mutations in these residues might, in fact, be conductive in a voltage dependent manner and only appear non-functional in traditional voltage-clamp conditions. It is also unclear whether the HIF1 mutation alters proton permeation or whether HIF-ASIC1a localized to mitochondria.*

We have examined the response to acid stimulation of CHO cells expressing HIF-ASIC1a held under a broad range of voltages and found no detectable acid-evoked change in currents at any voltage (please see new Figure 3). As a comparison, the WT channel responded to acid stimulation under all voltages. As for the proton permeability of HIF-ASIC1a, since we detected no acid-induced current, it is unlikely that this mutant permeated any ion at all. In terms of mitochondrion-localized HIF-ASIC1a, as we explained earlier (please see our response to major comment #2 above), mtASIC1a does not directly contribute to the acid-induced cell death.

*8) The authors find that co-administration of amiloride (unknown concentration) with pH 6.0 eliminates acid-gated currents, but does not affect acidosis-induced changes in cell viability (*Figure 3—figure supplement 1*). This seems in direct opposition to the results of Xiong et al., Cell 2004, (*Figures 4 and 5*). Similarly, the current manuscript suggested that pretreatment with amiloride for 1 hour did prevent acid-dependent changes in cell viability. Xiong et al., previously reported that incubating neurons for 10 min prior and during acidosis with amiloride prevented toxicity and amiloride incubation for extended periods of time is toxic (Xiong et al., Cell 2004). Additional information on the methodology as well as some inclusion of a discussion of the difference in these results should be provided.*

We have provided schematic diagrams for these experiments in new Figure 3—figure supplement 3. As the reviewers pointed out, Xiong et al. (50) included 10 min preincubation with amiloride to show a partial rescue effect on acid-induced neuronal death. This was not different from our observation that pretreatment with the drug was needed for neuroprotection. Since the previous study did not comment on the effect of co-application of amiloride with acid (50), our results cannot be considered to be in conflict with the previous one. Also, the previous study specifically indicated that 1-hr treatment with amiloride alone “did not affect baseline LDH release” and the toxic effect of the drug only appeared after up to 5 hr of incubation (50). In our hands, treatment with amiloride at pH 7.4 for 1 hr caused merely about 5% neuronal death. Therefore, our results on amiloride effects are quite consistent with that shown by Xiong et al. (50).

9) PcTx1 inhibits ASIC currents by pushing the channel into the steady state desensitized state. If steady state desensitization is still toxic, then PcTx1 must be attenuating acid induced neuronal death by preventing the “non-conductive” conformational transition which activates RIP1-mediated death in manner similar to that suggested for 1 hour amiloride. This should be addressed within the Discussion.

We agree with the reviewers on this. Most likely, PcTX1 binding, in addition to shifting steady-state desensitization, also prevented the conformational transition required for RIP1 activation by ASIC1a. This point has been incorporated in the Discussion (third paragraph).

10) The authors show that the CP1 region of the C-terminus is toxic to neurons in a manner that is independent of acidic pH and partially reversed by Nec-1. Given the non-specificity of Nec-1 and the basic nature of the included ASIC1-C-terminal fragment, it is important to further explore this toxicity. Is the effect dependent on RIP1 or ASIC1a? The importance of the TAT peptide should also be assessed as a control.

CP1 peptide caused cell death and phosphorylation of RIP1 in cultured mouse cortical neurons (please see new Figure 4). The neuronal death and RIP1 phosphorylation induced by CP1 peptide can be partially rescued by Nec-1 (please see new Figure 4). Therefore, the toxicity of CP1 depends on RIP1, at least partially. We also observed that CP1 peptide led to death of CHO cells (please see new Figure 4—figure supplement 1), which do not endogenously express ASIC1a, suggesting that CP1 toxicity is independent of ASIC1a. Importantly, under the same conditions, neither the TAT peptide (as a negative control) nor the CP2, CP3, or CP4 peptide induced cell death (please see new Figure 4), supporting the specificity of the CP1 peptide.

*11) The most direct evidence supporting the idea that the low-pH induced ASIC-dependent neuron death does not require the channel's ionic conduction is from*
Figure 3
*where the authors show that a non-conducting ASIC (HIF-ASIC1a) is as efficient as the wild-type in mediating pH6.0-induced cell death. However, this experiment was done in CHO fibroblast cells. The cell death is only ∼ 30% (∼20% higher than in the “blank” control), way below the ∼80% observed in neurons (*Figure 1*). At minimum, the authors should test the two constructs in the ASIC knockout neurons and test whether they have similar ability in restoring the pH-induced cell death.*

These experiments have been done using *Asic1*^*-/-*^ neurons and results support that HIF-ASIC1a mediates acidic neuronal death equivalently as the WT channel (please see new Figure 3, Figure 3—figure supplement 5).

12) The authors rely heavily on pharmacology. There is a general lack of discussion of why drugs were used at certain concentrations and applied for particular durations; how specific they are at those concentrations; and why the situation is thought to be the same between the various cells/tissues studied.

In addition to pharmacology, we used ASIC1a knockout mice, shRNA to knock down RIP1 expression, different ASIC constructs and channel mutations to demonstrate the conductance-independent, acid-induced ASIC1a- and RIP1-dependent acidotoxicity. We chose the drugs and concentrations carefully based on the literature. Adding justification for the choice of drugs and concentrations for each drug used in the paper is uncommon which may distract readers from the main thrust of the story.

13) Along the similar lines, the authors use the caspase inhibitor, zVAD-fmk, as a major part of their argument that cells are dying by necroptosis and not apoptosis. It would be useful to have a positive control for this agent (e.g., by showing that it blocks the effect of staurosporine).

We have used the staurosporine-induced neuronal apoptosis as a positive control and shown that zVAD-fmk effectively blocked the apoptotic neuronal death (please see new Figure 1—figure supplement 1). We also observed that z-VAD-fmk significantly blocked Geldanamycin (GA)-induced apoptotic neuronal death (please see new Figure 1—figure supplement 2).

*14) What pH is induced in the brain during MCAO? The authors should estimate this value and, if it is very different from 6.0 (the value tested in in vitro experiments), they should present additional evidence that the mechanism they worked out* in vitro *is operative* in vivo*.*

Various studies in the literature have documented the drop of pH in ischemic brain. During ischemia, the oxygen depletion leads to anaerobic glycolysis. Protons are accumulated due to the over-production of lactic acid and ATP hydrolysis, which cause pH value to fall to 6.5-6.0 in the ischemic brain (Siesjo et al., 1996; [50]). We have now stated pH range found in the ischemic brain (please see subsection “RIP1 is recruited to ASIC1a and phosphorylated in ischemic brain”).

15) There is insufficient demonstration of the necroptotic phenotype at the level of morphology. The authors provide EM images of cells but these are few and not quantified.

We have now included quantification of EM images in the figure legend of Figure 1.

References:

Chen X, Grunder S (2007) Permeating protons contribute to tachyphylaxis of the

acid-sensing ion channel (ASIC) 1a. J Physiol 579: 657–670.

Chu XP, Miesch J, Johnson M, Root L, Zhu XM, Chen D, Simon RP, Xiong ZG (2002)

Proton-gated channels in PC12 cells. J Neurophysiol 87: 2555–2561.

Hoagland EN, Sherwood TW, Lee KG, Walker CJ, Askwith CC (2010) Identification

of a calcium permeable human acid-sensing ion channel 1 transcript variant. J Biol Chem 285: 41852-41862.

Sherwood TW, Lee KG, Gormley MG, Askwith CC (2011) Heteromeric acid-sensing ion channels (ASICs) composed of ASIC2b and ASIC1a display novel channel properties and contribute to acidosis-induced neuronal death. J Neurosci: 31:9723-9734.

Siesjo BK, Katsura K, Kristian T (1996) Acidosis-related damage. Adv Neurol 71:

209-233; discussion 234-206.

Waldmann R, Champigny G, Bassilana F, Heurteaux C, Lazdunski M (1997) A

proton-gated cation channel involved in acid-sensing. Nature 386: 173-177

[Editors' note: further revisions were requested prior to acceptance, as described below.]

The reviewers acknowledge the strength of the ionic substitution (NMDG) and pore mutant experiments at making a role for Ca influx through ASICs appear highly unlikely for necroptosis. Nevertheless, they note that the novelty of complete Ca–independence is substantial and also deviates from previous work from your group, and therefore requires evidence not only that increased Ca levels are unnecessary for necroptosis but also that Ca levels do not actually increase (e.g., through another mechanism; if the mechanism is distinct, of course the case can still be made that Ca may not enter through the ASIC pore). Reviewer 1 suggests an imaging experiment in neurons (rather than just CHO cells); Reviewer 2, with the same underlying concerns, requests a discussion of possible bases for the difference between this and previous work.

Reviewer #1 (General assessment and major comments):

The authors have responded to the previous reviews with additional experiments and the manuscript has been strengthened. It will be an important manuscript for the field. The one significant omission that remains is calcium imaging in neurons. The authors should report ASIC1a-dependent calcium signal in neurons treated with Nec-1 and in neurons transfected with the different ASIC1a constructs used in the death assays. The authors now show that acid-induced calcium signaling is absent in transfected CHO cells (this has been previously reported by Samways et al., as stated by the authors) and that acid-induced cell death occurs in the absence of extracellular calcium in neurons. Yet, as the authors eloquently discuss, calcium permeability and acid-induced calcium fluctuations have been the focus of the field for years. The irrelevance of calcium is such a significant reversal of previous data published by the authors as well as almost all reviews of ASIC1a-dependent processes. This assertion should be backed by all commonly available methodologies. As the authors have previously reported imaging of ASIC1a-induced calcium signals in neurons (Gao et al., Neuron, 2005), these measures of calcium are conspicuously absent. Given the importance of these results for the field, they should also be included.

We thank the reviewer for bringing up this important point. Following your suggestion, we have examined acid-induced Ca^2+^ response in cultured neurons from WT and ASIC1a-KO mice. As shown in the new Figure 3—figure supplement 2 and consistent with the previous work (50; 16), application of the pH 6.0 solution to WT neurons elicited a robust transient increase in the Fura-2 ratio, indicative of acid-induced elevation in intracellular Ca^2+^ concentration ([Ca^2+^]_i_). Inclusion of a cocktail of blockers for ionotropic glutamate receptors and voltage-gated Na^+^ and Ca^2+^channels suppressed the acid-evoked response by about 50% and the remaining response was largely inhibited by PcTX1 (50 nM). Together with the lack of acid-evoked changes in Fura-2 ratio in ASIC1a-KO neurons under the same conditions, these results indicate that the native ASIC1a channels in mouse neurons mediate Ca^2+^ influx and the acid-evoked Ca^2+^ signals are further amplified through other channels, such as voltage-gated Ca^2+^ channels and glutamate receptors. We further demonstrated that transient transfection of WT or RC-ASIC1a, but not HIF-ASIC1a, restored acid-evoked Ca^2+^ signals in ASIC1a-KO neurons, consistent with the electrophysiological measurement of acid-induced currents in CHO cells for these constructs for their “channel functionality”.

Intriguingly, the Ca^2+^ response required fast focal perfusion (∼50 µl/min) of the acidic solution to the neurons. This perhaps is not too surprising because it has long been recognized that fast solution exchange is necessary in order to elicit the transient proton-evoked ASIC currents typically seen in the literatures. If the acidic solution was applied slowly, especially with whole chamber solution exchange, the currents either failed to develop or became shallow and small. This is entirely consistent with the rapid desensitization property of these channels as only when the majority of channels in the entire cell open simultaneously can robust currents develop. We reasoned that for acid-induced cell demise, neither the solution exchange in the experimental setting nor the manifestation of tissue acidosis needed to occur fast or in a synchronous fashion throughout the entire cells. Therefore, the strict conditions required to elicit acid-evoked whole-cell currents and the consequent [Ca^2+^]_i_ rise are not exactly compatible with that of acid-induced death. Indeed, when the pH 6.0 solution was applied at a slower rate (∼15 µl/min) to neurons, the [Ca^2+^]_i_ rise became very shallow or nearly undetectable (new Figure 3—figure supplement 2).Therefore, although WT ASIC1a channels indeed mediate Ca^2+^ influx in neurons, the Ca^2+^ signals arising from these channels are likely to be very small in response to the development of tissue acidosis. Given that the non-conducting HIF-ASIC1a mutant mediated acid-induced death without generating any Ca^2+^ signal in neurons even with the fast solution exchange (new Figure 3—figure supplement 2), we argue that the Ca^2+^ signals, no matter big or small, are not needed for the acid-induced neuronal death. We have now included these new data and revised the manuscript accordingly.

Reviewer #2 (Minor comments):

*One thing that still puzzles me, and perhaps other readers, is the nearly total lack of Ca*^*2+*^
*influx-dependence of low pH-induced cell death as observed in this paper; yet the*
[50]
*paper demonstrated clear Ca*^*2+*^*-dependence. The experimental conditions in the two studies appear to be similar. The authors did not address this apparent discrepancy as asked by the reviewers. They should at least include some discussion to address this.*

We now add new data showing that the pH 6.0 solution indeed evoked [Ca^2+^]_i_ rise in WT, but not ASIC1a-KO neurons (new Figure 3—figure supplement 2). We show that transient expression of WT and RC-ASIC1a, but not HIF-ASIC1a restored the Ca^2+^ response. However, as explained in our response to Reviewer 1, the Ca^2+^ response, just like the typical transiently lasting acid-induced currents, only occurred with the fast application of the acidic solution. At a slower perfusion rate, the Ca^2+^ response became very small or negligible. For cell death experiments, we washed the cells twice with the acidic solution in order to ensure a complete solution exchange in the dish (or well). Therefore, the rate of acid application was slow. We believe that tissue acidosis also occur slowly in the ischemic/reperfused brain. Therefore, the conditions required to observe robust acid-induced ASIC currents and the consequent [Ca^2+^]_i_ rise are not compatible with that for acid to induce neuronal death. The previous conclusion ion channel conductance and Ca^2+^ dependence of ASIC1a-mediated cell death were pretty much based on correlative data, which do not unequivocally prove the involvement of channel function per se in the induction of neuronal death. With respect to the 2004 paper (50), the “Ca^2+^-free” solution used actually contained 0.2 mM Ca^2+^, which could cause other complications that comprised the necroptotic death. We have now included this point in the Discussion of the revised manuscript.

[Editors' note: further revisions were requested prior to acceptance, as described below.]

The new experiments on the response of neurons to changes in pH raise the interesting possibility that the rate of solution application is the variable that accounts for the differences between the present and previous work. However, as pointed out by the reviewer, the information given in the manuscript suggests that the pH change in the slow-flow condition was so slow that it raises the concern that the cells were never exposed to the solution at all, which would explain the lack of Ca signal detected. The editors recognize that this perception may be a misunderstanding, owing to an incomplete description of the Y-tube apparatus and/or how it was verified that the solution reached the cell. If so, please edit the manuscript to make the methods clearer, so that this concern can be alleviated. If, however, no verification was done, it may be necessary either to provide alternative evidence to justify the claim that flow rate is the key variable determining the Ca signals, or to remove that conclusion and relegate the hypothesis of flow rate to the Discussion.

Reviewer #2:

*The authors addressed the concern of the reviewers by imaging low pH-induced Ca*^*2+*^
*changes in cortical neurons. They conclude that low pH perfusion leads to Ca*^*2+*^
*influx only when at high perfusion speed (50 µl/min) but not at low speed (15 µl/min). The authors further imply that the low speed, which induces no detectable Ca*^*2+*^
*increase, may be closer to the pathophysiological conditions. While I was generally satisfied with the previous version and did not specifically ask for additional measurements, I found the data added in this revision is totally unconvincing. The perfusion was done with focal perfusion using a Y-tube (opening size not specified) placed ∼ 500 µm away from the neuron. The concentration of the perfusion solution (i.e. the pH change) a neuron experiences is determined by the opening and the shape of the tube, and the angle at which the tube is aligned against the neuron. A simple explanation for the lack of Ca*^*2+*^
*change during slow perfusion could be that the neuron never experienced a significant pH change, and this had little to do with the speed of the pH change. After all, a15 µl/min perfusion against a dish (1 ml volume?) is very slow and there might be a large pH gradient between the tube opening and the cell body (even for a 500 µm distance). It's not clear to me why the authors did not use whole-dish perfusion (as done for the CHO cells) where the time it takes to change the whole bath can be more accurately measured/estimated. In summary, it appears to me that the conclusion that Ca*^*2+*^
*change is speed-dependent is meaningless, unless both the degree and the speed of pH change of bath surrounding the neuron are known.*

We appreciate the important point raised by the reviewer. Following the editor’s suggestion, we have provided more detailed description of the Y-tube apparatus (see subsection “Electrophysiological recordings”) and performed a validation experiment comparing currents of a non-desensitizing (or very slowly desensitizing) proton-gated channel, TRPV1, evoked by low pH at fast and slow perfusion rates. We show that in whole-cell recordings the proton-activated TRPV1 currents (either by pH 6.0 or pH 5.0) were not affected by the perfusion rate whereas the development of ASIC1a current (by pH 6.0) was severely hampered by the slow perfusion rate (Figure 3—figure supplement 3). These results demonstrate that in our system, the cells under study were in fact exposed equally to the acidic solution with both fast- and slow-flow rates and the differences observed between the fast and slow perfusion for Ca^2+^ and current responses of ASIC1a were due to the fast desensitization of this channel.

It is well-known that slow ligand application distorts the current of fast desensitizing ligand-gated channels, such as AMPA receptors. Although ASICs do not desensitize as fast as the AMPA receptors, we still routinely observe a delay and a strong suppression of the acid-induced ASIC current in whole chamber perfusion as well as focal perfusion at relatively slow flow rates. Therefore, just like the AMPA receptors, the proton-evoked ASIC currents can only be reliably recorded if the low pH solution was applied quickly to the cell. The Y-tube apparatus has allowed us to achieve this goal in imaging and electrophysiological assays in the past 20 years. The open end of the Y-tube has an inner diameter of ∼100 μm (Figure 6). Therefore, even though the values for flow rates (15 µl/min and 50 µl/min, adjusted by changing the height of the solution reservoir) appear to be small as compared to those typically reported for whole-chamber perfusion in the literatures, the distances and areas covered by the perfused solution can be quite far (∼30 mm/sec if the solution maintains a perfect cylindrical shape or at least 2 mm in diameter after 20 sec of continued perfusion if the solution merely dribbles out from the apparatus). Even with the most conserved estimate, if the 5 μl acid solution resulting from 20 sec application at the rate of 15 µl/min forms a perfect sphere, it would still occupy a space with a radius of 1080 μm (Figure 6, based on the formula (4/3)π r^3^), which is approximately 2 mm in diameter. This distance should be sufficient to cover the cells situated about 500 μm (0.5 mm) away from the tip of the Y-tube (Figure 6). In reality, the perfused solution most likely forms a cone-shaped stream rather than a sphere and the areas it covers are likely much bigger than the above estimate. Therefore, a 500 μm distance from the tip of the Y-tube is both sufficient for the solution to reach to the cell and ideal for avoiding any flush-induced cell shape distortion.

Author response image 1.The most conserved estimate of space occupied by the 20s-administration of a perfusion solution with Y-tube at the flow rate of 15 µl/min. The solution coming out from the Y-tube was assumed to form a perfect sphere, which represents a large underestimate of the area it would cover at the bottom of the dish.**DOI:**
http://dx.doi.org/10.7554/eLife.05682.022